# Ultra-fast photodetectors based on high-mobility indium gallium antimonide nanowires

Dapan Li[1,2], Changyong Lan [1,3], Arumugam Manikandan[4], SenPo Yip[1,2,5], Ziyao Zhou[1,2], Xiaoguang Liang[1,2], Lei Shu[1,2], Yu-Lun Chueh [4], Ning Han[6] & Johnny C. Ho [1,2,5]

Because of tunable bandgap and high carrier mobility, ternary III-V nanowires (NWs) have demonstrated enormous potential for advanced applications. However, the synthesis of large-scale and highly-crystalline $In_xGa_{1-x}Sb$ NWs is still a challenge. Here, we achieve high-density and crystalline stoichiometric $In_xGa_{1-x}Sb$ $(0.09 < x < 0.28)$ NWs on amorphous substrates with the uniform phase-purity and <110>-orientation via chemical vapor deposition. The as-prepared NWs show excellent electrical and optoelectronic characteristics, including the high hole mobility (i.e. $463\,cm^2\,V^{-1}\,s^{-1}$ for $In_{0.09}Ga_{0.91}Sb$ NWs) as well as broadband and ultrafast photoresponse over the visible and infrared optical communication region (1550 nm). Specifically, the $In_{0.28}Ga_{0.72}Sb$ NW device yields efficient rise and decay times down to 38 and 53 μs, respectively, along with the responsivity of $6000\,A\,W^{-1}$ and external quantum efficiency of $4.8 \times 10^6$ % towards 1550 nm regime. High-performance NW parallel-arrayed devices can also be fabricated to illustrate their large-scale device integrability for next-generation, ultrafast, high-responsivity and broadband photodetectors.

[1] Department of Materials Science and Engineering, City University of Hong Kong, Kowloon 999077, Hong Kong SAR. [2] Shenzhen Research Institute, City University of Hong Kong, 518057 Shenzhen, P.R. China. [3] School of Optoelectronic Science and Engineering, University of Electronic Science and Technology of China, 610054 Chengdu, P.R. China. [4] Department of Materials Science and Engineering, National Tsing Hua University, Hsinchu 30013, Taiwan. [5] State Key Laboratory of Terahertz and Millimeter Waves, City University of Hong Kong, Kowloon 999077, Hong Kong SAR. [6] State Key Laboratory of Multiphase Complex Systems, Institute of Process Engineering, Chinese Academy of Sciences, 100190 Beijing, P. R. China. Correspondence and requests for materials should be addressed to N.H. (email: nhan@ipe.ac.cn) or to J.C.H. (email: johnnyho@cityu.edu.hk)

Due to the extraordinarily high carrier mobility, III−V compound semiconductors are widely considered as promising candidates to replace Si for the next-generation electronics[1]. Especially, III-V nanomaterials, such as nanowires (NWs), have been evidently demonstrated with intriguing physical and chemical properties originating from their one-dimensional configuration[2]. In this regard, various methods have therefore been explored for the synthesis of high-quality III-V NWs, including chemical vapor deposition (CVD)[3–5], metal-organic CVD[6,7], laser ablation[8] and molecular beam epitaxy (MBE)[9], etc. Among many III−V materials, ternary compounds (e.g. InGaSb and InGaAs) have been always reported to yield the more superior electrical properties as compared to their binary counterparts[1,10–12]. For example, InGaSb gives the higher hole mobility than GaSb and InSb[12]. However, it is still a substantial technological challenge to achieve large-scale and highly crystalline ternary III−V NWs in a controllable and low-cost manner[13–15].

At the same time, III−V NW materials are often illustrated with excellent optoelectronic characteristics[2,13,16,17], which can be employed as active materials for photodetectors[18,19], lasers[20], and light emitting diodes[21]. Because of the tunable composition and tailorable bandgap, ternary III−V NWs are particularly advantageous as photosensing elements for broadband photodetectors[22–24]. In specific, InGaSb, which can stoichiometrically be expressed as $In_xGa_{1−x}Sb$, has a bandgap spanning from 0.17 eV ($x = 1$) to 0.72 eV ($x = 0$)[25], is an ideal alloy for tunable infrared (IR) detection. More importantly, since the NW geometry can yield enhanced photon absorption as well as provide efficient charge separation and carrier collection with marginal recombination loss, III-V NWs have always been shown with the impressive photodetection performance[16]. For instance, Liu et al. reported that InAs NW photodetectors showed excellent photodetection performance with high responsivity ($4.4 \times 10^3$ A W$^{−1}$), high external quantum efficiency (EQE, $1.03 \times 10^6$%), and high detectivity ($2.6 \times 10^{11}$ Jones)[26]. InGaAs NW photodetectors were also fabricated to give excellent performance with broadband response, high responsivity ($6.5 \times 10^3$ A W$^{−1}$) and large EQE ($5.04 \times 10^5$%)[27]. Nevertheless, to date, there is still very limited investigation on the photodetection behavior of InGaSb NWs owing to the lack of high-quality InGaSb NWs.

Herein, based on our previously developed enhanced CVD synthesis of ternary $In_xGa_{1−x}$As NWs[28,29], we report the successful growth of high-quality $In_xGa_{1−x}$Sb NWs with high hole mobility and efficient photosensing characteristics in both visible and short-wave (SW) IR regimes. Crystalline, high-mobility and <110>-oriented $In_xGa_{1−x}$Sb NWs with the uniform controllable In concentration, $x$, up to 28 at.%, can be readily achieved. Interestingly, a high hole mobility of 463 cm$^2$ V$^{−1}$ s$^{−1}$ is realized for $In_{0.09}Ga_{0.91}$Sb NWs, indicating the high quality of the ternary NWs. Furthermore, the ternary NWs show sensitive photodetection performance in the IR optical communication region (1550 nm). For the optimized $In_{0.28}Ga_{0.72}$Sb NW devices, they show broadband photodetection performance at room temperature, with a responsivity of 6000 A W$^{−1}$, specific detectivity of $3 \times 10^9$ Jones, external quantum efficiency (EQE) of $4.8 \times 10^6$% and response time of 38 μs under IR illumination (1550 nm) at a bias voltage of 2 V. This efficient response in the tens of microsecond range represents one of the fastest responses among all NW IR photodetectors reported in the literature. Importantly, these NWs can as well be fabricated into NW parallel arrays-based devices, demonstrating their technological potential as active materials for next-generation, ultrafast and efficient room-temperature photodetectors.

## Results

**Growth and characterization of $In_xGa_{1−x}$Sb nanowires.** $In_xGa_{1−x}$Sb NWs are synthesized by the two-step solid-source CVD method[28]. In order to synthesize $In_xGa_{1−x}$Sb NWs with different chemical stoichiometry of In and Ga, InSb and GaSb powders in various mixture ratios (e.g. 10:1, 20:1, 30:1 and 40:1 in wt.%) are used as the precursor source. The chemical composition of the obtained NWs could then be evaluated by using energy dispersive X-ray spectroscopy (EDS), where they are found to be $In_{0.15}Ga_{0.85}$Sb, $In_{0.22}Ga_{0.78}$Sb, $In_{0.09}Ga_{0.91}$Sb, and $In_{0.28}Ga_{0.72}$Sb for the source mixture ratio (InSb:GaSb in wt.%) of 10:1, 20:1, 30:1, and 40:1, respectively (details can be seen in Supplementary Tables 1, 2 and Supplementary Fig. 1). This slender variation of the NW composition indicates the robustness of the present CVD method in controlling the NW chemical stoichiometry, which is beneficial for assessing the NW composition-dependent properties. Furthermore, as shown in the scanning electron microscope (SEM) image of typical $In_{0.28}Ga_{0.72}$Sb NWs (Fig. 1a), smooth and clean NWs with a length greater than 10 μm are grown on the amorphous SiO$_2$/Si substrate. When the In concentration of the NWs is increased, the NW morphology is maintained, but there is a slight change on their growth density due to the increasing amount of InSb powder in the precursor source mixture (Supplementary Fig. 2). Based on the transmission electron microscope (TEM) characterization, the average diameter of $In_{0.15}Ga_{0.85}$Sb, $In_{0.22}Ga_{0.78}$Sb, $In_{0.09}Ga_{0.91}$Sb, and $In_{0.28}Ga_{0.72}$Sb NWs are found to be very similar, with the values of $60 \pm 10$, $45 \pm 10$, $60 \pm 5$, and $57 \pm 10$ nm, accordingly, extracted from the statistics of more than 20 NWs for each sample group. Evidently, there is not any significant effect on the NW morphology and diameter when the In concentration of NWs is varied. Simultaneously, elemental mapping of the NW is also performed as presented in Fig. 1b−f. It can be seen that the chemical constituents of In, Ga and Sb are distributed uniformly along both axial and radial directions of the NW, indicating the excellent NW composition homogeneity obtained in this simple growth. Notably, Au is only detected in the spherical tip region of the NW, which further confirms the catalytic vapor−liquid−solid mechanism employed here.

In addition, the morphology and growth orientation of the NWs could also be assessed by detailed TEM studies. As depicted in the TEM image of a typical $In_{0.28}Ga_{0.72}$Sb NW in Fig. 2a, the NW body has a uniform diameter along the entire length of NW without any noticeable tapering behavior. Based on the corresponding selected area electron diffraction pattern (Fig. 2a inset), the distinctive and sharp diffraction spots clearly suggest the excellent crystallinity of the NW with the zinc blende (ZB) structure and growth direction of <110>. This particular growth orientation is as well observed for the rest of other NWs, being independent of the NW diameter and NW composition (Fig. 2b and Supplementary Fig. 3), which is in perfect agreement with other III−V NW materials reported since their <110> growth direction always consists of the relatively low surface energy planes for the effective NW nucleation and formation[4,30]. Almost all of the obtained NWs exhibit a substantial tapering effect right next to their catalytic tips, where the NW diameters increase gradually from the tip to the NW body along the neck-bending region with a length of 10−50 nm (Fig. 2c, d and Supplementary Figs. 4−6). It is also noted that all the catalytic tips consist of the typical hexagonal $Au_{10}In_3$ structure (PDF Card: 42-0821) with a lattice spacing of 0.22 nm along the {04$\bar{4}$0} planes (Supplementary Fig. 7, Supplementary Table 3), while all catalyst/NW interface orientations are identified as $Au_{10}In_3$ (04$\bar{4}$0)| $In_xGa_{1−x}$Sb (110) for the NWs growing along the <110> direction (Fig. 2d and Supplementary Figs. 4−6, 8). It is still controversial whether the

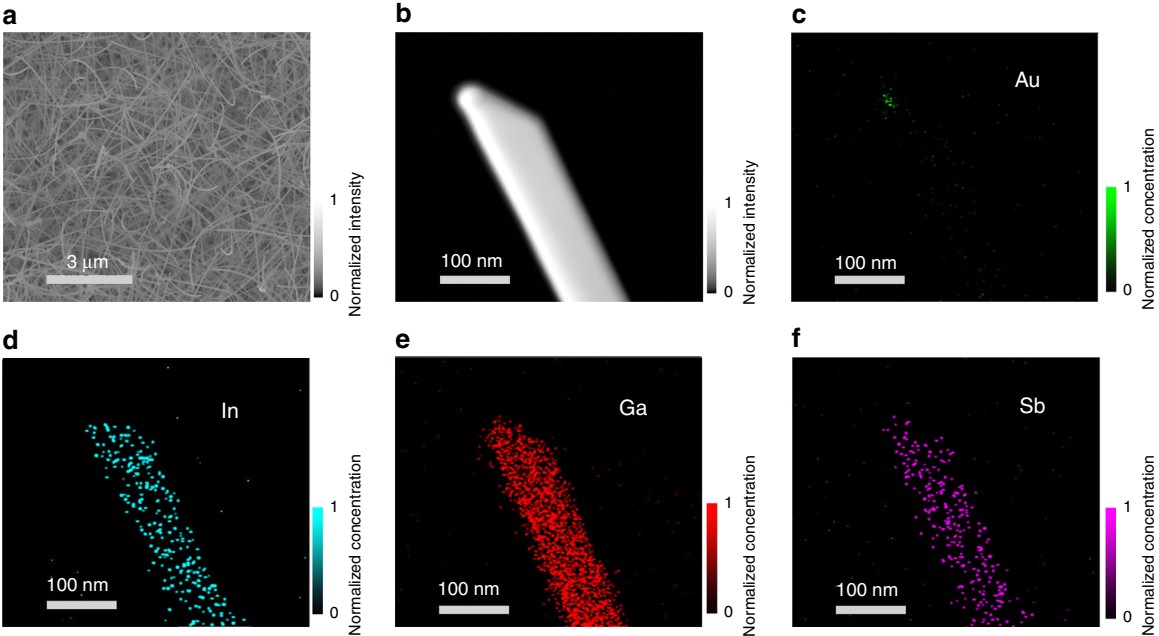

**Fig. 1** Morphology and chemical composition of $In_xGa_{1-x}Sb$ nanowires. **a** Scanning electron microscopy image of $In_{0.28}Ga_{0.72}Sb$ nanowires (NWs) prepared by using the precursor powder mixing ratio of InSb:GaSb = 40:1 (wt.) and 0.1-nm-thick Au films as the catalyst. **b** Transmission electron microscopy image and (**c**–**f**) energy dispersive X-ray spectroscopy mapping of a typical $In_{0.28}Ga_{0.72}Sb$ NW. The gray scales denote the measured intensity. The color scales stand for the normalized concentration of Au, In, Ga and Sb measured, while the value of 1 represents the maximum concentration of each element detected, being 74.6 atomic % of Au for the NW tip, 14.2 atomic % of In, 38.7 atomic % of Ga and 47.1 atomic % of Sb for the NW body, respectively

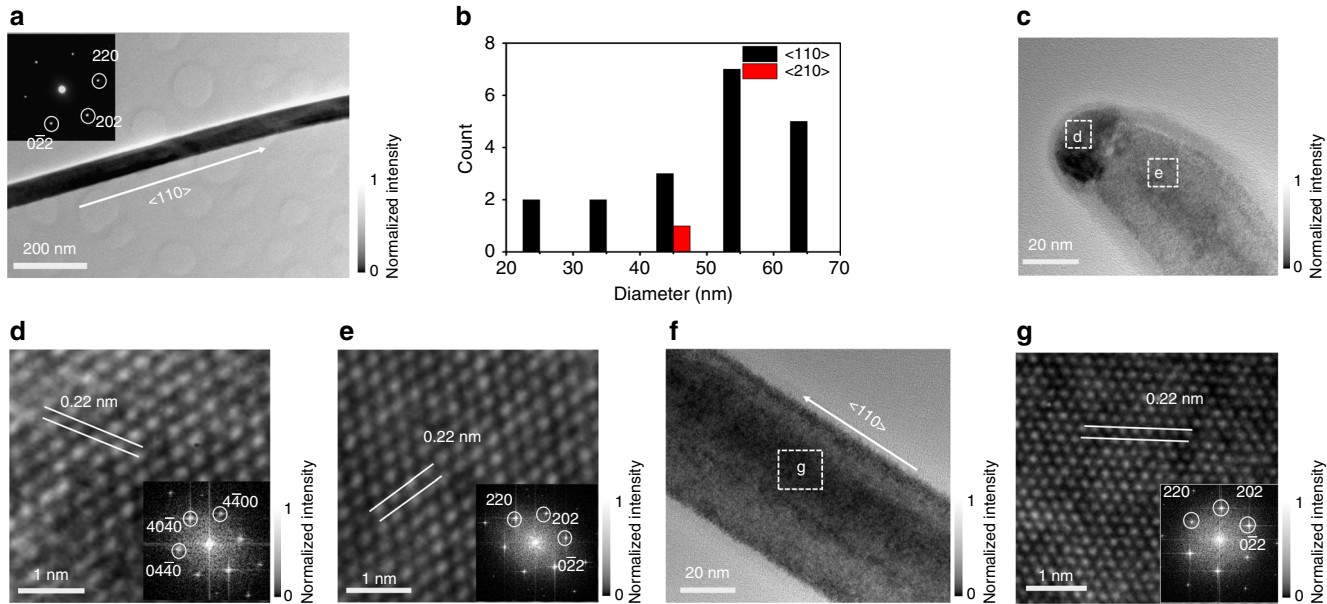

**Fig. 2** Growth orientations and crystallinity of $In_{0.28}Ga_{0.72}Sb$ nanowires. **a** Transmission electron microscopy (TEM) image of a representative nanowire (NW). Inset shows the corresponding selected area electron diffraction pattern and growth orientation of the NW. **b** Statistics of the NW growth direction as a function of the diameter. **c** High-resolution TEM image of the tip region of the NW imaged in panel (**a**). **d** Lattice fringes of the tip region of the NW. Inset displays the corresponding fast Fourier transform (FFT) pattern. **e** Lattice fringes of the neck-bending region of the NW. Inset shows the corresponding FFT pattern. **f** High-resolution TEM image of the body region of the NW imaged in panel (**a**). **g** Lattice fringes of the body region of the NW. Inset displays the corresponding FFT pattern. The gray scales denote the measured intensity

epitaxial relation between the NW and the catalyst particle is caused by a quasi-vapor−solid−solid process or vapor−liquid−solid (VLS) process[31]. Nevertheless, since the In concentration is only varied in a relatively small range (0 to 28 at.%), all NWs consist of the similar lattice spacing of 0.22 nm along {110} planes

without any noticeable defects existed in the NW bodies (Fig. 2e, f and Supplementary Figs. 4−6, 9, 10). All these results indicate the uniform phase- and orientation-purity as well as the excellent crystallinity of our $In_xGa_{1-x}Sb$ NWs for further device utilizations. Furthermore, reflectance spectra were taken to evaluate the

optical properties of the NWs (Supplementary Fig. 11). As anticipated, the bandgap of the NWs is found to decrease with the increasing In content (Supplementary Fig. 12), indicating the effective incorporation of In into the GaSb lattice as well as the good composition uniformity of the NWs.

**Growth mechanism of $In_xGa_{1-x}Sb$ nanowires.** In order to shed light onto the NW growth mechanism, especially investigating the formation of NW neck-bending regions, TEM and EDS line scan are carefully performed on the tip region of a typical $In_{0.28}Ga_{0.72}Sb$ NW along its axial direction (Fig. 3a, b). It is found that catalytic tip is mainly composed of the elements of In and Au, which is perfectly consistent with the observation of $Au_{10}In_3$ seed in the above discussion. Notably, there is no Ga and Sb elements witnessed in the tip; the Ga and Sb concentration start to be detected at the tip/NW interface. Both Ga and Sb concentration are gradually increased along the length of the NW neck, and eventually saturated at the NW body when the neck region ends. Based on these observations, the growth mechanism of $In_xGa_{1-x}Sb$ NWs can be proposed as depicted in Fig. 3c: Firstly, during the growth, Au droplets are first formed as

catalysts on the substrate. Secondly, when the growth proceeds and the precursor vapor arrives at the substrate, AuIn alloy seeds would be established by the diffusion of In atoms into the Au catalyst droplets. Since the affinity of In in Au nanoparticles (NPs) is much higher as compared with the one of Ga, binary AuIn alloy seeds are formed here instead of the ternary AuInGa ones[31]. Next, with the continuous feeding of In species, In will be saturated in the AuIn droplets, leading to the deposition of In metal. In the end, the deposited In will then react with Ga and Sb species, forming $In_xGa_{1-x}Sb$ NWs. With the continuous feeding of Ga and Sb species, the Ga and Sb will reach a stabilized value. In this case, for the continuous deposition of In from AuIn droplets, the section (i.e. neck-bending region) adjacent to the catalyst droplets would become In rich with a tapered morphology due to the difference between reaction kinetics and constituent diffusion of Ga and Sb with In, which leads to gradient distribution of Ga and Sb along the axial direction[32]. The qualitative understanding of the growth steps can not only help to assess the formation mechanism of this ternary $In_xGa_{1-x}Sb$ NW material system, which is notoriously difficult to obtain, but also potentially guide us to further control the growth of NW with enhanced crystallinity and property. It should also be pointed out

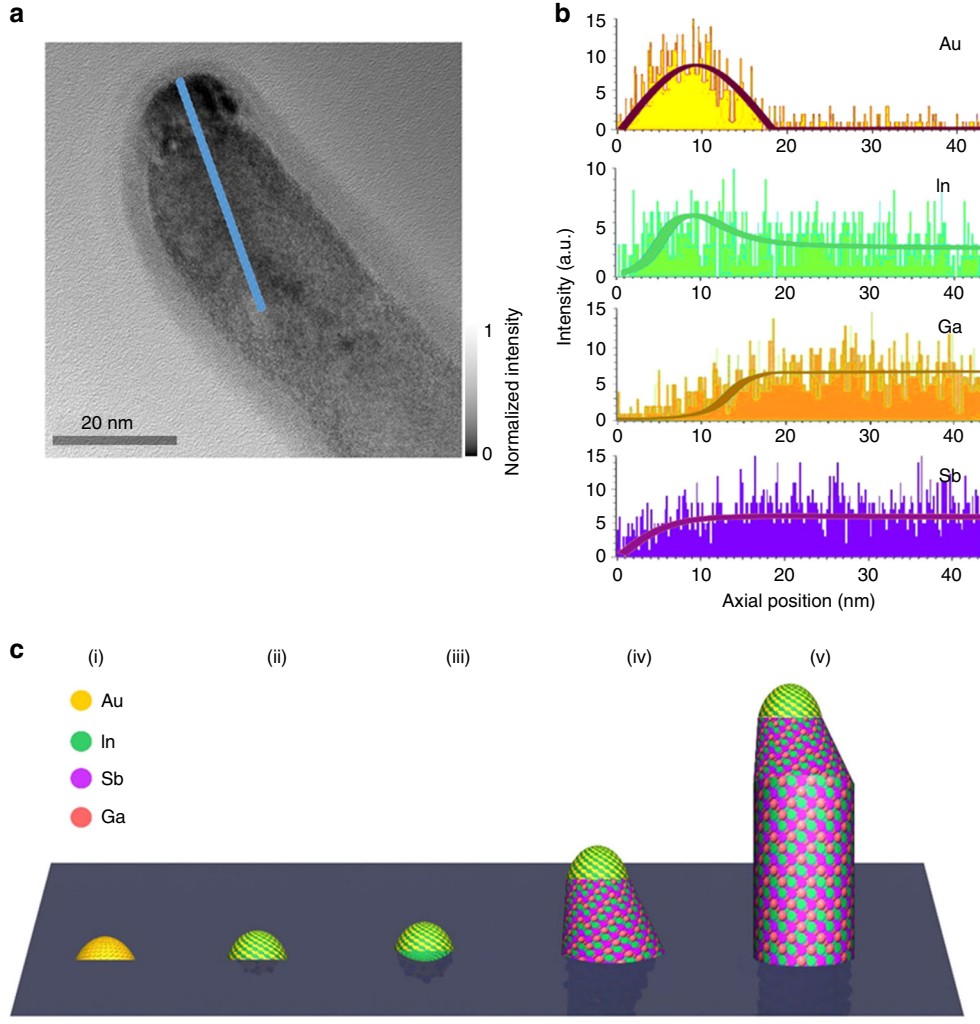

**Fig. 3** Proposed growth mechanism of $In_xGa_{1-x}Sb$ nanowires. **a** Transmission electron microscopy image of the tip region of a typical $In_{0.28}Ga_{0.72}Sb$ nanowire (NW). The gray scale denotes the measured intensity. **b** Energy-dispersive X-ray spectroscopy line scan across the NW tip, neck and body in the axial direction. The location of the line scan is indicated by the blue colored line in panel (**a**). **c** Illustrative schematics of four different proposed growth steps involved during the NW growth. They are: (i) formation of the gold (Au) nanoparticle, (ii) formation of the AuIn alloy, (iii) In precipitation from the AuIn alloy, (iv) formation of $In_xGa_{1-x}Sb$ neck with the increasing Ga concentration, and (v) growth of $In_xGa_{1-x}Sb$ NW with the steady Ga concentration

that the growth of ternary $In_xGa_{1-x}Sb$ NWs is different from the one of GaSb NWs as reported in our previous work[5], where AuGa alloy droplets were employed as the catalyst. Furthermore, the ternary NWs show a smaller diameter as compared with typical GaSb NWs. The addition of In can affect the NW growth dynamics while the uncontrolled radical NW growth may be constrained.

**Electrical properties of $In_xGa_{1-x}Sb$ nanowires.** Apart from the NW formation mechanism, it is also essential to evaluate the electrical properties of the obtained $In_xGa_{1-x}Sb$ NWs. In this case, global back-gated NW field-effect transistors (FETs) were fabricated with Ni as the source-drain electrodes. From the transfer ($I_{ds} - V_{gs}$) and output ($I_{ds} - V_{ds}$) characteristic curves (Fig. 4a–c) of a representative single $In_{0.09}Ga_{0.91}Sb$ NW FET, respectively, it is clear that the drain-source current ($I_{ds}$) decreases with the increase of gate voltage ($V_{gs}$), indicating the typical p-type conducting behavior of the NW. In order to evaluate the contact quality of NW devices, their contact resistance were measured by fabricating single NW devices with multiple channel length (Supplementary Fig. 13). It can be seen that the contact resistance of the $In_{0.09}Ga_{0.91}Sb$ NW device is found to be 11.3 kΩ, which is ten times smaller than the resistance of the NW body. Combined with the linear relationship as observed in their output characteristics, the Ohmic-like contact between the NWs and the electrodes can be inferred. This relatively small contact resistance means that there is only a small voltage drop across the electrical contact. In this case, the mobility estimated from the transfer characteristics can indeed represent the actual mobility values of the NW devices. Importantly, the device also exhibits an impressive ON/OFF current ratio of $10^5$

for $V_{ds} = 0.4$ V (Fig. 4b). The field-effect hole mobility ($\mu$) can be then extracted from the transfer curves utilizing the following equations[33]:

$$\mu = g_m \cdot \frac{L^2}{C_{ox}} \cdot \frac{1}{V_{ds}}, \qquad (1)$$

where $g_m$ is the transconductance, defined as $\partial I_{ds}/\partial V_{gs}$, $C_{ox}$ is the gate capacitance and $L$ is the channel length. $C_{ox}$ can be obtained from the finite element method by using COMSOL MultiPhysics software (Supplementary Fig. 14). For a typical $In_{0.09}Ga_{0.91}Sb$ NW transistor, when $L$ is 4.2 μm, NW diameter is 41 nm, gate capacitance is determined to be 0.26 fF for 50 nm $SiO_2$ dielectric layer from COMSOL and peak transconductance is found to be $7 \times 10^{-8}$ S for $V_{ds} = 0.1$ V (Supplementary Fig. 15), then the peak $\mu$ of the NW device can be calculated as high as 463 $cm^2 V^{-1} s^{-1}$ (Fig. 4d). This mobility value is higher than the one of pure GaSb NWs[4], indicating the superior electrical properties of ternary III−V NWs, consistent with the behavior observed in bulk materials[1,10–12]. Furthermore, the hole mobility is close to metal-oxide-semiconductor field-effect transistor (MOSFET) based on MBE synthesized epitaxial InGaSb film (~510 $cm^2 V^{-1} s^{-1}$)[10], further confirming the high quality of the NWs. As shown in the Fig. 4e, this hole mobility value is also compared and contrasted with other maximum mobility values reported for III−V NWs in the literature. In order to further assess the mobility distribution of $In_xGa_{1-x}Sb$ NWs with different In concentration, more than 40 NW devices were measured for each In concentration and the corresponding statistics of the extracted mobility is presented in Fig. 4f. Explicitly, the mobility distribution can be well fitted by Gauss function, while the average mobility decreases with the increase of In concentration. The change of band structure and

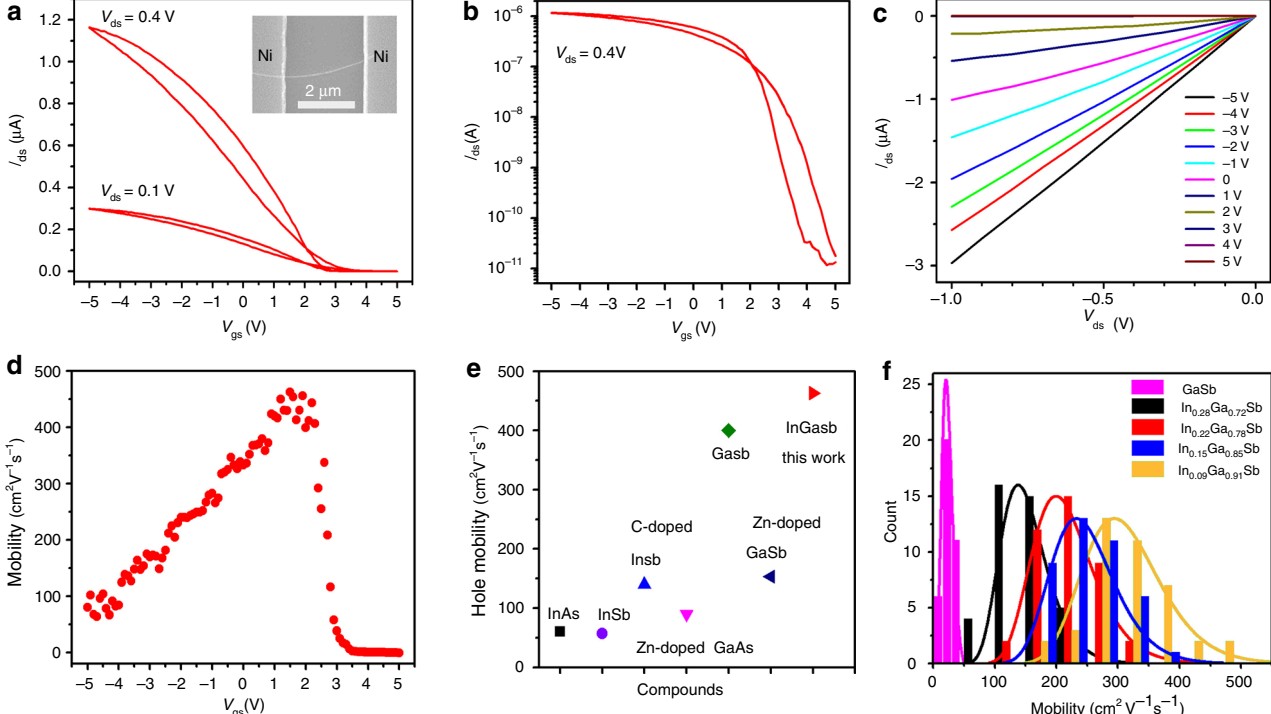

**Fig. 4** Electrical characterization of $In_xGa_{1-x}Sb$ nanowire transistors. **a−c)** Transfer and output characteristics of a representative nanowire (NW) field-effect transistor (FET) made of single $In_{0.09}Ga_{0.91}Sb$ NW channel. Inset shows the scanning electron microscopy image of the back-gated NW FET with Ni metal contacts. **d** Mobility assessment of the NW FET under source-drain bias of 0.1 V. **e** Hole mobility of different semiconductor NW FETs. The maximum hole mobility of InAs, InSb, C-doped InSb, Zn-doped GaAs, GaSb, and Zn-doped GaSb is adapted from refs. [4,43–47]. Their average hole mobility values are also compiled in Supplementary Fig. 15c. **f** Statistics of the extracted mobility of ~40 NW FETs with different NW stoichiometries as the device channel

increased carrier scattering by random distribution of In in the lattice may be the reason for the reduced hole mobility with the increase of In concentration[25]. The slight variation of both In and Ga concentration (Supplementary Table 2) in each NW sample groups would give the additional variation to the corresponding mobility values, contributing to the wider mobility distribution as compared to pure GaSb NW devices. To further exclude any effect of surface modification (e.g. adsorbents and passivation) on the NW device mobility, the typical $In_{0.09}Ga_{0.91}Sb$ NW device was measured under different conditions, including vacuum and surface passivation with $(NH_4)_2S$, in which the results suggest that high device mobility values here are indeed attributed to the intrinsic NW properties, instead of relating to any other extrinsic effects (e.g. surface modification, Supplementary Figs. 17 and 18). Based on both Fig. 4c and Supplementary Fig. 15c, it is demonstrated that the mobility values of our $In_{0.09}Ga_{0.91}Sb$ NW devices, regardless of the maximum and average values, are the highest among many others of III−V NW devices reported to date. For comparison, GaSb NWs were also synthesized with the similar method here and their corresponding mobility values were measured with an average value of only 26 cm$^2$ V$^{-1}$ s$^{-1}$. This relatively low mobility value could be attributed to the lattice defects, such as stacking faults and inversion domains, observed in the GaSb NW (Supplementary Fig. 16), which is in a distinct contrast to the high crystallinity of $In_xGa_{1-x}Sb$ NWs (Fig. 2 and Supplementary Figs. 4−6). The introduction of In in GaSb NWs not only reduces the defect concentration but also leads to the favorable changes in their electrical device performance. The

device mobility values of $In_xGa_{1-x}Sb$ NWs are therefore much improved as compared to the pure GaSb NW ones with the NW channel grown by the similar technique. Also, this variation trend of the mobility as a function of In content is consistent with the previous literature that the hole mobility first increases with the introduction of In, and then decreases after it reaches the peak value[12]. In any case, the high hole mobility of those NWs makes them suitable for many advanced electronic applications.

**Photodetection performance in the visible regime.** As III−V materials are excellent optoelectronic materials, it is interesting to explore the photodetection properties of $In_xGa_{1-x}Sb$ NWs. As shown in the output curves of the $In_{0.28}Ga_{0.72}Sb$ NW device with different light intensities of 532 nm (Fig. 5a), it is clear that the illuminated output current increases with the increase of light intensity under a constant source-drain bias ($V_{ds} = 2$ V) and a zero gate bias ($V_{gs} = 0$ V). This illuminated current is typically known as the photocurrent ($I_{ph}$), which is defined as the current difference between the illuminated state and the dark state. The photocurrent is as well compiled as a function of the light intensity (Fig. 5b), in which the dependence can be fitted by the following sub-linear relationship:

$$I_{ph} = A\Phi^{\alpha}, \tag{2}$$

where $A$ and $\alpha$ are the fitting parameters and $\Phi$ is the light intensity. Through fitting, we can obtain the value of $\alpha$ as 0.6.

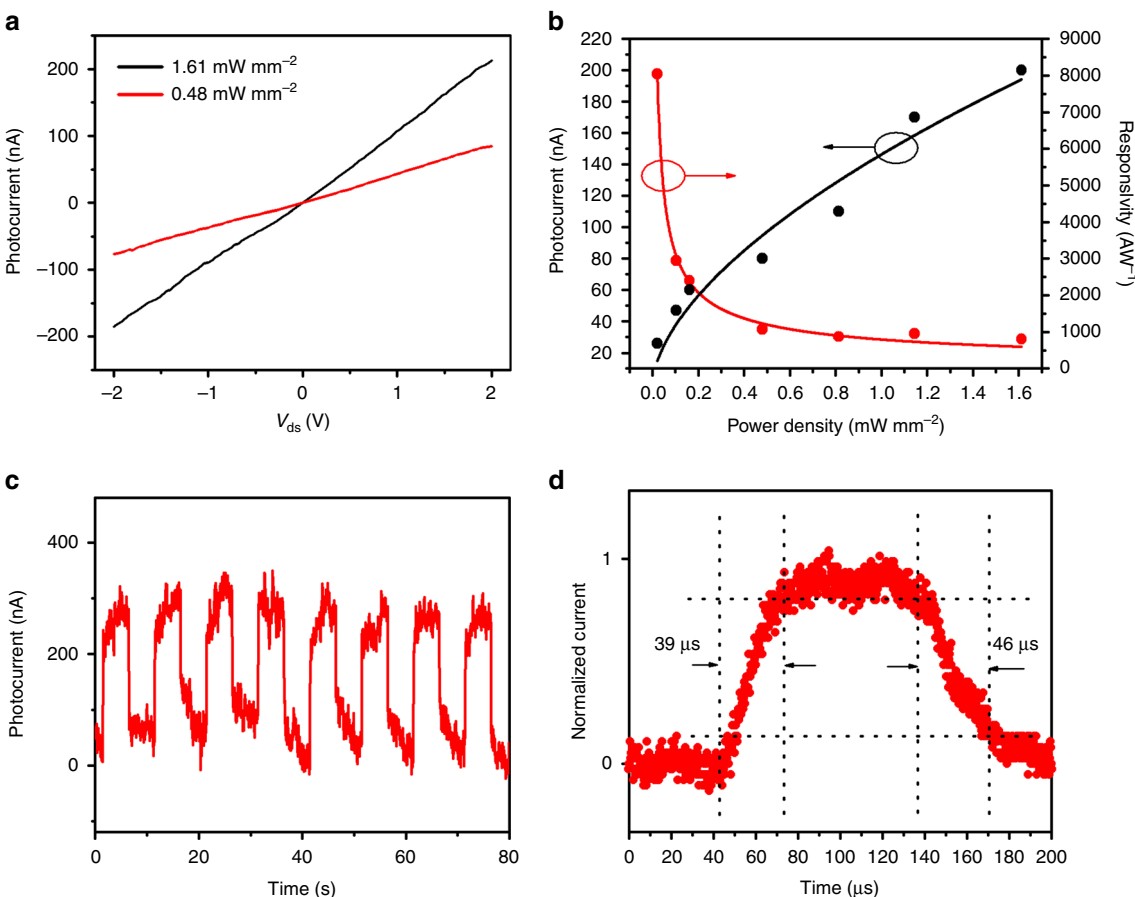

**Fig. 5** Photodetection of single $In_{0.28}Ga_{0.72}Sb$ nanowire (532 nm laser). **a** Current–voltage curves under the illumination intensities of 0.5 and 1.6 mW mm$^{-2}$, respectively. **b** Photocurrent and responsivity as a function of the incident illumination intensity. **c** Photoresponse of the nanowire photodetector under the illumination intensity of 1.6 mW mm$^{-2}$. The chopped frequency is 0.1 Hz. **d** A high-resolution transient photoresponse of the device to illustrate the rise time and decay time constants. In panels (**b**−**d**), the gate bias is 0 V and the source-drain bias is 2 V

Notably, this sublinear relationship between photocurrent and light intensity is always observed in NW-based photodetectors, which originates from the complex processes of electron-hole generation, trapping and recombination in the NW[34–36]. On the other hand, photoresponsivity is another important parameter to assess the photodetector performance, which can be expressed as:

$$R = \frac{I_{ph}}{\Phi S}, \tag{3}$$

where $S$ is the active area of the photodetector. According to Eqs. (2) and (3), $R$ should be proportional to $\Phi^{\alpha-1}$, which is shown in Fig. 5b. The device shows a very large responsivity (more than 8000 A W$^{-1}$) when the light intensity is set to 0.02 mW mm$^{-2}$, illustrating the good sensitivity of the NW detector. Meanwhile, the response speed and stability of the photodetector under modulated light intensity are also critical for practical utilization. The time-resolved photoresponse measurements were performed by periodically modulating the illumination (i.e. chopping on and off) as given in Fig. 5c, d. The efficient switching characteristics indicate the admirable periodicity and stability of the NW detector. The rise and decay time constants, defined as the time interval for the current rise from 10 to 90 % of the peak value and vice versa[37], can be determined from the high-resolution photoresponse measurement (Fig. 5d). This way, the rise and decay time constants are found to be 39 and 46 μs, respectively, indicating the fast response of the device. More importantly, the In$_{0.28}$Ga$_{0.72}$Sb NW photodetectors likewise yield the similar photoresponse against the illumination of both 405 and 635 nm (Supplementary Figs. 19 and 20), in which all these results strongly designate their remarkable broadband detection and

spectral response in the visible regime. Photodetection performance of the NWs with other In contents were also measured and the results are shown in Supplementary Fig. 21 and Supplementary Table 4. It can be seen that there is not any significant difference in their performances among different NW compositions. Typically, the photocurrent is proportional to the product of carrier mobility and carrier life-time ($I_{ph} \propto \mu\tau$)[38]; however, the density of different types of recombination centers can drastically change the value of carrier lifetimes here[38]. As a result, it is anticipated that those NWs would exhibit the similar photodetection performance although their carrier mobilities are different.

**Photodetection performance in the infrared regime.** At the same time, the cost-effective and efficient IR photodetection at room temperature is still technological challenging due to the lack of appropriate active photosensing materials. Here, our In$_x$Ga$_{1-x}$Sb NW photodetectors are demonstrated with the efficient photoresponse characteristics towards the short-wave IR regime (i.e. optical communication region at 1550 nm). As shown in the typical output characteristic with different illumination intensities of 1550 nm in Fig. 6a, the single In$_{0.28}$Ga$_{0.72}$Sb NW photodetector exhibits the increase of photocurrent with the increase of the illumination intensity at $V_{ds} = 2$ V and $V_{gs} = 0$ V. Figure 6a inset even shows the excellent switching behavior of the device with various incident intensities. It is observed that the IR response of those NW devices would enhance substantially when the In concentration of the NW device channel is increased accordingly, which can probably be attributed to the reduced bandgap as well as the enhanced absorption coefficient towards 1550 nm irradiation (Supplementary Figs. 11, 22 and

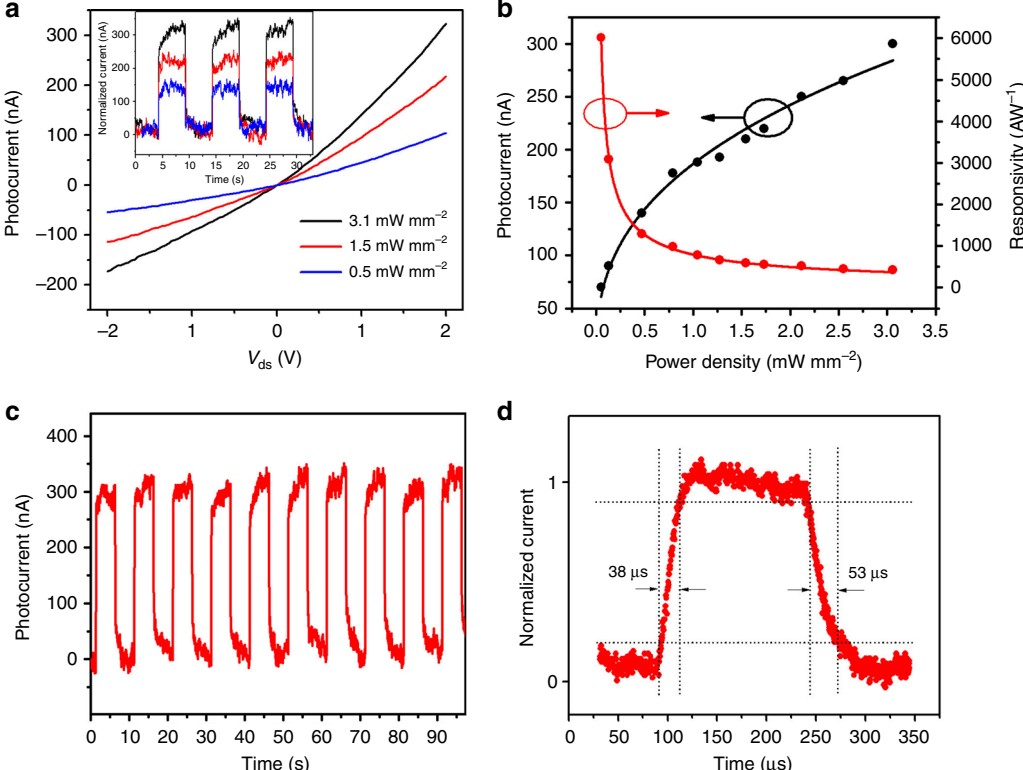

**Fig. 6** Photodetection of single In$_{0.28}$Ga$_{0.72}$Sb nanowire (1550 nm laser). **a** Current–voltage curves under the illumination intensity of 3.1, 1.5, and 0.5 mW mm$^{-2}$, respectively. Inset shows their corresponding photoresponse under different illumination intensities. **b** Photocurrent and responsivity as a function of the incident illumination intensity. **c** Photoresponse of the nanowire photodetector under the illumination intensity of 3.1 mW mm$^{-2}$. The chopped frequency is 0.1 Hz. **d** A high-resolution transient photoresponse of the device to illustrate the rise time and decay time constants. In panel (**a**) inset, (**b**–**d**), the gate bias is 0 V and the source-drain bias is 2 V

**Table 1 Performance metrics of single nanowire photodetectors**

| Material | Incident wavelength (nm) | Responsivity (A W$^{-1}$) | Operating temperature | Dark current/ Bias voltage | External quantum efficiency (%) | Response speed | Ref. |
|---|---|---|---|---|---|---|---|
| InP | 532 | 6.8 | RT | ~0.3 nA/−5 V | $1.6 \times 10^3$ | — | 48 |
| InAs | 532 | — | RT | — | — | 3.12 s | 18 |
| InAs | 532 | — | RT | ~40 µA/−1 V | — | — | 49 |
| InGaSb | 532 | 8120 | RT | 6 µA/2 V | $6.5 \times 10^6$ | 39 µs | This work |
| InP | 700–1000 | — | RT | ~125 pA /10 V | — | — | 50 |
| GaAs | 400–1200 | — | RT | — | — | — | 51 |
| InAs | 632–1470 | $5.3 \times 10^3$ | RT | ~1 nA/2 V | — | — | 52 |
| InAsP | 1240–1910 | — | 5 K | — | — | — | 53 |
| InAsP | 700–3500 | 5417 | RT | 0.2 µA/0.5 V | $3.95 \times 10^5$ | — | 22 |
| InGaAs | 1100–2000 | $6.5 \times 10^3$ | RT | 144 nA/0.5 V | $5.04 \times 10^5$ | 280 ms | 27 |
| GaAsSb | 1100–1660 | 2.37 | 77K–RT | <100 nA/0.1 V | — | — | 54 |
| GaAsSb | 1160–1550 | $1.7 \times 10^3$ | RT | ~200 nA/1 V | $1.62 \times 10^5$ | 60 ms | 55 |
| InGaSb | 1550 | $6 \times 10^3$ | RT | 1.9 µA/2 V | $4.8 \times 10^6$ | 38 µs | This work |

Supplementary Table 5)[27]. To further understand the photosensing characteristic of these NW detectors, the dependence of device photocurrent on different illumination intensities is measured and depicted in Fig. 6b. Similarly, all the measured data can be well fitted by the Eqs. (2) and (3) for the determination of $\alpha$ and responsivity values. The $\alpha$ value is found to be 0.7 for all biases. Based on these dependences, several other key performance parameters, such as external quantum efficiency (EQE) and specific detectivity (D*), of the NW device can also be assessed. Generally, these parameters can be defined as:

$$\text{EQE} = R\frac{hc}{e\lambda}, \qquad (4)$$

$$D* = R \cdot \sqrt{\frac{S}{2eI_{\text{dark}}}}, \qquad (5)$$

where $h$ is the Planck's constant, $c$ is the velocity of light, $e$ is the absolute charge of an electron, $\lambda$ is the wavelength of incident light and $I_{\text{dark}}$ is the dark current. In this case, at a low incident intensity of 0.05 mW mm$^{-2}$ and $V_{\text{ds}} = 2$ V, the values of $R$, EQE and $D*$ are determined to be 6000 A W$^{-1}$, $4.8 \times 10^6$% and $3.7 \times 10^9$ Jones, respectively. These measured performance parameters are already comparable to those of the state-of-the-art III−V NW photodetectors as shown in Table 1.

In addition, the good repeatability and fast response time are also two important requirements for advanced photodetectors. Figure 6c gives the photocurrent response of the single In$_{0.28}$Ga$_{0.72}$Sb NW photodetector under the light intensity of 3.1 mW mm$^{-2}$ with a chopping frequency of 0.1 Hz. It can be seen that the device on- and off-state can be effectively modulated by the chopped illumination, indicating the robustness and excellent stability of the device. The device on- and off-state currents for each of ten cycles shown here vary within 300 ± 50 and 25 ± 20 nA, respectively, within the noise level, which suggests stable optical reversibility over the measured time interval. The possible origins of the observed fluctuations could be due to the absorption/desorption of surface molecules. The photocurrent values reveal that no pumping or priming effects need to be taken into account for the explored time scale. A high-resolution transient curve is measured to precisely determine the rise and decay time constants of the studied device (Fig. 6d). Here, the rise and decay time constants can be identified as 38 and 53 µs, accordingly. This efficient response in the tens of microsecond range represents one of the fastest responses among all NW IR photodetectors reported in the literature (Table 1).

Further reduction of the surface trap concentration by surface passivation with (NH$_4$)$_2$S can decrease the response time down to 24 µs as shown in Supplementary Fig. 23. Since there are abundant surface states in nanostructured materials, these surface states are typically first saturated by photo-induced carriers to yield the slow response times. In this case, the small changes of the response times (i.e. with and without any surface passivation) observed can infer the relatively high crystal quality of NWs. This sensitive photodetection performance of In$_x$Ga$_{1−x}$Sb NW devices can be attributed to their enhanced surface-to-volume ratio of the one-dimensional NW channel, homogenous NW composition and crystallinity as well as appropriate controllable bandgap, etc.[39]

In order to further demonstrate the feasibility of utilizing these NW photodetectors for the large-scale device integration, we employ the well-established NW contact printing and the same device fabrication procedure to achieve In$_x$Ga$_{1−x}$Sb NW parallel array FETs with efficient photoresponse characteristics in the 1550 nm regime. As shown in Supplementary Fig. 24, at a low incident light intensity of 0.05 mW mm$^{-2}$ and $V_{\text{ds}} = 2$ V, the values of $R$ and $D*$ are found to be about 4500 A W$^{-1}$ and $5 \times 10^8$ Jones, respectively. As compared with the single NW photodetector, these $R$ and $D*$ values give a slight reduction, which is probably due to the improper alignment of the NW arrays, NW-to-NW variations, breakage of NWs as well as nonoptimized electrical contact between NWs and electrodes[40]. Anyway, ultrafast response times, ranging between 30 and 50 µs, are still observed for these NW parallel array devices. In the future, the photodetector performance can be further improved through the optimization of the NW parallel array density, alignment and channel length scaling, etc.

## Discussion

High density and crystalline In$_x$Ga$_{1−x}$Sb NWs on amorphous substrates with <110>-orientation are successfully synthesized via two-step CVD method. The high quality of those NWs is confirmed by the high hole mobility of the fabricated NW devices. Specifically, In$_{0.09}$Ga$_{0.91}$Sb NWs show a peak mobility of 463 cm$^2$ V$^{-1}$ s$^{-1}$, which is one of the highest values among all III−V NWs and comparable to InGaSb MOSFET, suggesting the potential of these NWs for advanced electronics applications. Furthermore, when configured into photoconductive detectors, the InGaSb NWs show the sensitive photodetection performance in both visible and IR regime. In particular, the single In$_{0.28}$Ga$_{0.72}$Sb NW photodetector can yield the excellent responsivity of 6000 A W$^{-1}$, EQE of $4.8 \times 10^6$% and specific detectivity of $3.7 \times 10^9$ Jones at

1550 nm, together with the ultrafast response time down to 38 μs. Notably, the gain of a photoconductor can be expressed as:

$$G = \tau/t_r, \qquad (6)$$

where $\tau$ is the life-time of photo-generated free carriers and $t_r$ is the transit time of carriers between two electrodes. Due to the large surface-to-volume ratio, NWs exhibit a high density of surface states. Therefore, the Fermi energy of NWs is typically pinned at the surface forming a depletion layer, which results in the effective electrons and holes separation and thus longer life-time of photo-generated free carriers[41]. Owing to the high carrier mobility of InGaSb NWs, the transit time is also observed to be short. This way, the photoconductors made from InGaSb NWs would have a high gain. On the other hand, the response time ($\tau_o$) is dominantly determined by the life-time of photo-generated free carriers, and is expressed as $\tau_o = (1 + p_t/p)\tau$, where $p_t$ is the trapped carrier density, $p$ is photo-generated free carrier density[38]. As the responsivity is expressed as:

$$R = e \cdot \frac{\eta\lambda}{hc} \cdot G, \qquad (7)$$

where $\eta$ is the quantum efficiency (the fraction of the incident optical power that contributes to electron−hole pair generation, which is widely used in photoconductive detectors)[42]. The life-time of photo-generated free carriers can be estimated from Eqs. (6) and (7), where it can be rewritten as $\tau = R\frac{hc}{\eta\lambda} \cdot \frac{1}{e} \cdot t_r$. In this work, the transit time is found to be $\sim 5 \times 10^{-11}$ s, while the photo-generated free carrier life-time can be determined as 0.24/$\eta$ μs by taking all the known values into Eqs. (6) and (7) at 1550 nm. If the quantum efficiency is 1 without any traps, the life-time should be 0.24 μs, which is contradictory to our observed response times. However, the quantum efficiency is known to be far smaller than 1 because of the reflectance of light, insufficient absorption of light attributable to the finite thickness of NW and anisotropic absorption of light arising from the one-dimensional geometry, etc. Also, there are many carrier traps for NWs due to the large surface-to-volume ratio of NWs. As a result, it is reasonable to obtain the response time on the order of tens of microseconds here. Furthermore, high-performance and large-scale NW parallel array devices have also been fabricated to further illustrate the technological potential for industrialization. All these results evidently reveal these In$_x$Ga$_{1-x}$Sb NWs having enormous potential for the next-generation, ultrafast, high-responsivity and broadband photodetectors over visible and short-wave IR regions.

## Methods

**Synthesis of In$_x$Ga$_{1-x}$Sb NWs.** The In$_x$Ga$_{1-x}$Sb NWs were synthesized by a two-step solid-source CVD method in a two-heating-zone horizontal furnace. To get In$_x$Ga$_{1-x}$Sb NWs with different chemical stoichiometry, InSb and GaSb mixture powders with different weight ratios (10:1, 20:1, 30:1, and 40:1 in wt.%) were used as the solid-source. For the NW growth, 1 g of the source powder was first loaded in a boron nitride crucible, which was placed in the center of the quartz tube in the upstream zone. Then, Si/SiO$_2$ (50-nm-thick thermally grown oxide) wafer pieces coated with 0.1 nm (nominal thickness) Au film as the catalyst was used as the growth substrate, which was placed in the center of the quartz tube in the downstream zone. The subsequent growth temperatures, both at upstream and downstream zones, as well as the flow rate of hydrogen carrier gas, were next carefully adjusted in order to control the properties of NWs. For example, under a weight ratio of 40:1, the pressure inside the quartz tube was evacuated down to a base pressure of $1.5 \times 10^{-3}$ Torr by a mechanical pump. After that, 100 sccm H$_2$ gas was introduced into the quartz tube for 20 min. Then, both upstream and downstream zones were heated to 750 and 510 °C, respectively, in 10 min, which were kept for a duration of 180 min. Eventually, the furnace was cooled naturally to room temperature. During the synthesis, the pressure in the quartz tube was kept at 2.1 Torr. The detailed growth parameters of all components are also shown in Supplementary Table 1.

**Material characterization.** The morphologies of the as-grown NWs were examined with a field emission SEM (XL30, Philips, Netherlands) and a TEM (CM-20

Philips, Netherlands). Crystal structures of the NWs were determined by HRTEM imaging (JEOL 2100F and JEM-3000F FEGTEM, 300 kV, JEOL Co., Ltd., Tokyo, Japan). Elemental mappings were performed using EDS detectors attached to the JEOL 2100F and FEI Tecnai G$^2$ F30 to measure all the chemical composition of the as-grown NWs. The chemical state of the as-prepared NWs was examined by XPS (model 5802, ULVAC-PHI, Inc., Japan). For the TEM and elemental mappings, the NWs were first suspended in anhydrous ethanol solution by ultra-sonication and drop-casted onto the TEM grid for the corresponding characterization.

**NW FET and photodetector fabrication and measurement.** NW FETs and photodetectors were fabricated by drop-casting the NW suspension onto degenerately boron-doped p-type Si substrates with a 50-nm-thick thermally grown silicon dioxide layer. Photolithography was utilized to define the source and drain regions, and 50-nm-thick Ni film was thermally deposited as the contact electrodes followed by a lift-off process. The electrical performance of the fabricated back-gated FETs was then characterized with a standard electrical probe station and an Agilent 4155C semiconductor analyzer (Agilent Technologies, Santa Clara, CA). For photodetector characterizations, the 405, 532, 635 and 1550 nm lasers were used as the light sources. A power meter (PM400, Thorlabs, USA) was used to calibrate the power of the incident light. For the 405, 532 and 635 nm lasers, a home-made mechanical chopper and an attenuator was used to modulate the light and tune the power of the light, respectively. The modulator (AFG 2005, Arbitrary Function Generator, Good Will Instrument Co. Ltd) attached to the 1550 nm laser was used to modulate and tune the power of the IR irradiation. For determining the response time of the photodetector, a low noise current amplifier (SR570, Stanford Research Systems, USA) combined with a digital oscillator (TBS 1102B EDU, Tektronix, USA) was used to get high-resolution current versus time curves.

## Data availability

The data that support the findings of this study are available from the corresponding authors upon reasonable request.

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

## Acknowledgements

This research was financially supported by the General Research Fund (CityU 11204614) and the Theme-based Research Scheme (T42-103/16-N) of the Research Grants Council of Hong Kong SAR, China, the National Natural Science Foundation of China (51672229 and 61605024), the Science Technology and Innovation Committee of Shenzhen Municipality (Grant JCYJ20170818095520778) and a grant from Shenzhen Research Institute, City University of Hong Kong.

## Author contributions

J.C.H. and N.H. conceived the project. J.C.H, N.H., D.L. and C.L. prepared the manuscript. D.L. and S.P.Y. grew the NWs. S.P.Y. implemented the XRD measurement. Z.Z. and X.L. performed the SEM while A.M. and Y.-L.C. carried out the TEM characterization and analysis. D.L. and L.S. fabricated all the NW devices. D.L., C.L and Z.Z. performed all the electrical measurements and data analyses. All authors examined and commented on the manuscript.
