## [Peer Review File · Nature Communications]

Reviewers' comments:

Reviewer #1 (Remarks to the Author):

Dear Authors,

This is a well written and interesting manuscript describing the synthesis of InGaSb nanowires and their electrical and optical characterization. The topic is interesting and worth publication as quite good performance in terms of hole mobility and optical properties are reported. However, I would not recommend publication in "Nature Communications" due to several reasons described below.

1) The manuscript contains a detailed characterization and discussion about the growth mechanism and the nucleation. Although required for the completeness of the manuscript, the topic is far from new. The conclusions are very similar to the discussions for instance related to the VLS growth of InAs/GaSb nanowire heterostructures developed over the last decade. Unfortunately this reviewer did not find any new aspect in the presentation.

2) One main argument for the study is that InGaSb nanowires are expected to have a higher hole mobility than GaSb. Indeed high values are reported. However, the data in figure 4d is contradictory, the mobility is the highest for the lowest In content. Unfortunately, the GaSb reference is not clear as it is taken from a different study. Reading the manuscript it is not clear if really the introduction of In is beneficial for the transport?

3) It is claimed that the contacts are ohmic. However, no systematic study is presented. What is the access resistance in these devices and how does it affect the mobility determination?

4) The transistors show a hysteresis. How is this taken into account when determining the mobility?

5) It is claimed that the transistors have an I_{on}/I_{off} ratio of 10^5 at $V_{ds}=0.4V$. However no supporting data for this very bold claim is provided!

6) The optical data is interesting, but the response times are very long and the applied biases high. The authors need to provide a much more detailed discussion on the possible surface effects and provide data for passivated nanowires, to make sure that this is not a material quality problem.

In general my recommendation to the editor and the authors is to transfer the manuscript to a more appropriate journal.

Reviewer #2 (Remarks to the Author):

This manuscript reports the synthesis of ternary $In_xGa_{1-x}Sb$ nanowires (NWs) with different In compositions by a two-step chemical vapor deposition method. The NWs exhibit good crystallinity, high mobility, fast and high photoresponse. $In_xGa_{1-x}Sb$ ternary nanowire growth and devices have not been reported before. The results from this work are interesting and could be significant, however not convincing enough in the current form, since some important issues/questions have not been well presented, discussed and understood. I do not think the manuscript is of sufficient high quality for publication in high impact journals such as Nature Communication. My specific questions and comments are:

1) This work reports composition controllable InGaSb NWs achieved by mixing different ratio of binary material powder sources. However the results are not consistent. For powder source ratio of 30:1 (the third highest InSb case), the lowest In concentration (9%) was obtained. The supporting information mentions that it "is probably due to the lower substrate employed for the optimized

growth". This is not clear and a more detailed explanation should be provided. In any case, this does not show a good and reproducible control of the NW growth/composition. Also, the paper claims that "... as shown in the typical SEM image of In_{0.28}Ga_{0.72}Sb NWs (Figure 1a), straight, smooth and dense NWs with the length greater than 10 μm are non-epitaxially grown on the amorphous SiO₂/Si substrate. When the In concentration of the NWs is increased, the NW morphology is maintained with only the slight change in their growth density". The NWs are clearly not straight and also there is an obvious density variation. Especially for the In_{0.15}Ga_{0.85}Sb NWs, the density is obviously lower. The description in the paper should be made more accurate. In fact some morphology change with different In concentration is expected and should be discussed with regard to the growth conditions.

2) The chemical compositions of the NWs are obtained by EDS. The NW crystal structures are studied by TEM with very similar results. Surprisingly there is no mentioning of any defects such as stacking faults, for any of the NWs. I wonder how thorough the NWs were examined? The authors should clarify this. To get a better understanding of the NW properties and their corresponding device performance, optical spectroscopy such as photoluminescence should also be performed to measure the bandgap as well as the optical quality of the NWs.

3) In this work, a very high hole mobility of 463 cm²/Vs has been obtained. However for mobility calculation, the authors didn't show how to calculate the gate capacitance of the NW and what are the various parameters used. The Reference [5] cited didn't show the calculation either. In Table I the work by other researchers normally use 300 nm SiO₂ and metallic cylinder model to calculate gate capacitance, the work only use 50nm SiO₂ - it is important that it provides sufficient information to confirm the calculation and results.

4) It is shown that the 9% InGaSb NWs have the highest mobility and the results are presented in Fig. 4 (c) and (d). However Fig. 4(a) and (b) present the measurements of NW FET made of single In_{0.28}Ga_{0.72}Sb. Then the inset of Fig. 4(a) shows the SEM of In_{0.09}Ga_{0.91}Sb NW. It is confusing and the paper should just provide all the mobility measurement results from In_{0.09}Ga_{0.91}Sb NW FET to avoid confusion.

5) For the NW photodetectors, despite that impressive performance has been achieved, there is not much comparison (among different type of NWs), discussion and explanation. For example, compared with visible, a much faster response has been obtained for IR detection. Why? Compared with the In_{0.28}Ga_{0.72}Sb NWs, the In_{0.09}Ga_{0.91}Sb NW seems to have much lower noise and larger photocurrent (Fig. S11). Why the paper claims that In_{0.28}Ga_{0.72}Sb is the optimized device especially when the In_{0.09}Ga_{0.91}Sb NWs have higher mobility? How about the other two types of NWs with intermediate In compositions? How do they behave (in terms of responsivity, response time and detectivity) and why? To better understand the photodetector performance of the NWs with different In concentration (and thus bandgap), the dark current and spectral response curves should also be provided to fully understand the device performance.

6) For the photodetectors, the photocurrents sometimes are shown in nA and sometimes are shown in μA. I suppose for the former the dark current has been subtracted? For the comparison, it is better to show all the results (except for the response time measurements) under the same illumination intensity with dark current subtracted.

7) P. 22, it is mentioned that "the internal quantum efficiency is known to be far smaller than 1 because of the reflectance of light, insufficient absorption of light attributable to the finite thickness of NW and anisotropic absorption of light arising from the one-dimensional geometry". This is not correct. Internal quantum efficiency (IQE) is defined as the ratio of the number of charge carriers that are generated to the number of photons that are absorbed. It is mainly related to the material quality rather than the reflectance of the light etc. The description above is more relevant to EQE.

I also suggest the following minor corrections:

1) Fig. 1(b) is not an EDS image, it is an SEM image.

2) Please give the full name of "TEM" in Page 6, "NP" in Page 10. In Page 3, please add (MBE) after "molecular beam epitaxy" since MBE is used in the text afterwards.

3) In Fig. 3a, better to use the same colors and order for Au, In, Sb, Ga as Fig. 3b. The current colors and order is confusing.

- 4) Please provide a reference for Equation 1 and also the values for V_{gs} , g_m , C_{ox} , and L .
- 5) For Equation 3, the definition of area "S" should be given.
- 6) In the manuscript, when discussing the photodetection performance, the single NW device is often referred as single NW FET. This could be confusing and I suggest to call it as single NW photodetector.
- 7) Page 19, "In addition, the good repeatability and fast response speed are also two important requirement for advanced photodetectors", "requirement" should be "requirements".

Reviewer #3 (Remarks to the Author):

In this work, the authors investigated the high-performance transistors and photodetectors based on ternary III-V InGaSb nanowires. Indeed, III-V nanowires are promising optoelectronic materials; however, the reliable synthesis of InGaSb nanowires with controlled composition, crystallinity and morphology is still a big challenge till now. Here, using the enhanced chemical vapor deposition, high-density and crystalline stoichiometric InGaSb nanowires can be readily obtained on amorphous substrates, in particular with the excellent hole mobility and ultrafast photoresponse over both visible and infrared optical communication regions when these nanowires are configured into devices. Therefore, it is recommended to consider all these novel findings, which would contribute significant advancement to the scientific community. In any case, there are some questions needed to be addressed.

1. The authors mentioned that the mobility distribution can be well fitted by Gauss function, however, no Gauss fitting was made in the manuscript.
2. The information given in Table S1 is confusing, especially the percentage combustion being inconsistent with the values given in the table. For example, for the ratio 20:1, the mass of source is 1 g, and source loss is 0.26 g; in this case, the combustion percentage should be 26%, instead of 25.1% given in the table. The authors should make all the calculation more explicit. Furthermore, it is not suitable to use the term "combustion percentage" because the source is not combusted during the synthesis.
3. Similarly, for Table S2, why doesn't the molar ratio of In in the nanowires simply increase with the increasing of mass ratio of InSb powder? This reason would be important for the controllable synthesis of ternary III-V nanowires.
4. Regarding the nanowire array photodetectors, the reduced R and D^* should not arise from the higher parasitic capacitance. The capacitance can only affect the response speed of a photodetector. While the speed of the nanowire array photodetector is almost the same with single nanowire device as presented by the authors, this indicates that there is not any higher parasitic capacitance existed for the nanowire array devices.

Response to Reviewers' Comments and Suggestions on Manuscript NCOMMS-18-11449

We appreciate the referees for considering our manuscript and providing valuable comments. Accordingly, changes have been made in the manuscript, highlighted with red color. Below is our response to the reviewers' comments.

Response to the Reviewers' comments:

Reviewer #1:

1. The manuscript contains a detailed characterization and discussion about the growth mechanism and the nucleation. Although required for the completeness of the manuscript, the topic is far from new. The conclusions are very similar to the discussions for instance related to the VLS growth of InAs/GaSb nanowire heterostructures developed over the last decade. Unfortunately this reviewer did not find any new aspect in the presentation.

Response:

- We thank for the comment. As pointed out by the reviewer, it is essential to evaluate and understand the detailed growth mechanism of ternary InGaSb NWs for the completeness of this work. In fact, the growth mechanism presented here is very different from the typical VLS growth of InAs/GaSb NWs (B. M. Borg, *et al*, *Nano Lett.*, 2010, 10, 4080). In specific, for the growth of GaSb with 5% In (*i.e.* InGaSb) NW segment on the InAs NW stem, the formation of In-rich AuInGa alloy catalyst particle was found to inhibit the growth of InGaSb NW segments because of the thermodynamic reasons; therefore, it is challenging to achieve the uniform growth of InGaSb NW segments utilizing the ternary AuInGa based catalyst particles. On the other hand, in our experiments, we found that the binary AuIn catalyst particle can effectively catalyze the growth of ternary InGaSb NWs due to the employment of a higher growth temperature. This observed phenomenon is somewhat similar with the growth of InGaAs nanowhiskers (D. Sudfeld, *et al*, *Phase Transitions*, 2006, 79, 727), where the AuIn catalyst particle was also used there. As a result, all these new findings would further the advance of uniform ternary NW growth for practical device applications.

2. One main argument for the study is that InGaSb nanowires are expected to have a higher hole mobility than GaSb. Indeed high values are reported. However, the data in figure 4d is contradictory, the mobility is the highest for the lowest In content. Unfortunately, the GaSb reference is not clear as it is taken from a different study. Reading the manuscript it is not clear if really the introduction of In is beneficial for the transport?

Response:

- We appreciate for the valuable input. We have grown the GaSb NWs using the similar method as presented in this work and measured their corresponding field-effect mobility values as shown in **Figure S16**. It is clear that the average NW diameter increases slightly due to the uncontrolled radial growth and their corresponding hole mobility becomes smaller with an average value of only $26 \text{ cm}^2 \text{ V}^{-1} \text{ s}^{-1}$, being consistent with the previous literature (Yang *et al.*, *ACS Appl. Mater. Interfaces*, 2013, 5, 10946). Also, the obtained result agrees very well with the theoretical study on InGaSb films (Nainani *et al.*, *J. Appl. Phys.*, 2012, 111, 103706), in which the hole mobility increases with the introduction of In. After the mobility reaches a peak value, the mobility would then decrease accordingly. As a result, it is confirmed that adding In

is beneficial for enhancing the transport properties of GaSb NWs.

- In order to reinforce this viewpoint, we have added the following discussion: “For comparison, GaSb NWs were also synthesized with the similar method here and their corresponding mobility values were measured with an average value of only $26 \text{ cm}^2\text{V}^{-1}\text{s}^{-1}$ (Supporting Information Figure S16). This variation trend of the mobility as a function of In content is also consistent with the previous literature that the hole mobility first increases with the introduction of In, and then decreases after it reaches the peak value.¹²” to page 15 of the revised manuscript. We also modified Figure 4d to include the mobility data of GaSb NW devices.

3. It is claimed that the contacts are ohmic. However, no systematic study is presented. What is the access resistance in these devices and how does it affect the mobility determination?

Response:

- We thank for the valuable suggestion. We have measured the contact resistance of the NW devices by fabricating single NW devices with the multiple channel length. In specific, the contact resistance of $\text{In}_{0.09}\text{Ga}_{0.91}\text{Sb}$ NWs is found to be $11.3 \text{ k}\Omega$, which is 10 times smaller than the resistance of the NW body. Combined with the linear relationship as observed in their output characteristics, we can infer that the contacts are Ohm-like. This relatively small contact resistance means that there is only a small voltage drop on the contact. In this case, the mobility estimated from the transfer curves can indeed represent the actual mobility values of the NW devices.
- In order to illustrate all these arguments, the following discussion: “In order to evaluate the contact quality of NW devices, their contact resistance were measured by fabricating single NW devices with the multiple channel length (Supporting Information Figure S13). It is obvious that the contact resistance of the $\text{In}_{0.09}\text{Ga}_{0.91}\text{Sb}$ NW device is found to be $11.3 \text{ k}\Omega$, which is 10 times smaller than the resistance of the NW body. Combined with the linear relationship as observed in their output characteristics, the Ohm-like contact between the NWs and the electrodes can be inferred. This relatively small contact resistance means that there is only a small voltage drop on the electrical contact. In this case, the mobility estimated from the transfer characteristics can indeed represent the actual mobility values of the NW devices.” is added to page 13 of the revised manuscript.

4. The transistors show a hysteresis. How is this taken into account when determining the mobility?

Response:

- We thank for the comment. Usually, for NW transistors, the hysteresis is originated from the surface charge trapping, which can reduce the mobility of NW devices. Also, the mobility values obtained from the positive-to-negative gate sweep is always smaller than that from the reverse sweep due to the surface nature of NWs. In this work, in order to get the appropriate estimation of the mobility values, we employed the positive-to-negative gate sweep for all the parameter calculation. For example, the peak mobility of the typical $\text{In}_{0.09}\text{Ga}_{0.91}\text{Sb}$ NW FET is found to be 463 and $491 \text{ cm}^2\text{V}^{-1}\text{s}^{-1}$, for the positive-to-negative and the negative-to-positive gate sweeps, respectively (Figure R1). Although there is hysteresis observed, it is rather small, and we used the lower bound value of $463 \text{ cm}^2/\text{Vs}$ as its mobility in our analysis. In order to reinforce this argument, we have added Figure S15 to the revised Supporting Information.

Figure R1. Mobility determination of the typical $\text{In}_{0.09}\text{Ga}_{0.91}\text{Sb}$ NW FET with both positive-to-negative and negative-to-positive gate sweeps.

5. It is claimed that the transistors have an $I_{\text{on}}/I_{\text{off}}$ ratio of 10^5 at $V_{\text{ds}}=0.4\text{V}$. However no supporting data for this very bold claim is provided!

Response:

- We thank for the input. We have added the corresponding log plot of the transfer curve of the NW device as shown in Figure 4a inset in page 12 of the revised manuscript. It is clear that the $I_{\text{on}}/I_{\text{off}}$ ratio of $\sim 10^5$ is observed at $V_{\text{ds}} = 0.4\text{V}$.

6. The optical data is interesting, but the response times are very long and the applied biases high. The authors need to provide a much more detailed discussion on the possible surface effects and provided data for passivated nanowires, to make sure that this is not a material quality problem.

Response:

- We appreciate for this valuable comment. Actually, the applied biases are consistent with the ones widely utilized in the previous works, while the response times are very short as compared with the values reported in the literature (Table 1). In order to evaluate the surface effect, we also measured the photoresponse of passivated NW devices. In specific, the effect of surface passivation on the $\text{In}_{0.28}\text{Ga}_{0.72}\text{Sb}$ NW photodetector with the channel passivated with $(\text{NH}_4)_2\text{S}$ was thoroughly evaluated (Supporting Information Figure S21). It is found that the rise and decay time constants of the passivated InGaSb NW device were reduced from 38 and 53 μs to 24 and 37 μs , respectively, after surface passivation. These reduced response times can be explained by the minimized surface trap concentration owing to the effective surface passivation. Since the difference of the observed response times is relatively small as compared between with and without the surface passivation, all the discussion presented in this work would be based on the results without any surface passivation. In any case, the optical response can be confirmed due to the intrinsic properties of our InGaSb NWs, instead of the material quality issue.
- In order to reinforce this viewpoint, the following discussion: “In the meanwhile, further reducing the surface trap concentration by surface passivation with $(\text{NH}_4)_2\text{S}$ can decrease the response time down to 24 μs as shown in Supporting Information Figures S21. Although there are abundant surface states in the nanostructured materials, in which these states should be first saturated by the photo-induced carriers making the response times slower, these small changes

of the response time (i.e. with and without any surface passivation) can infer the relatively high NW crystal quality here.” is added to page 21 of the revised manuscript.

Reviewer #2:

1. This work reports composition controllable InGaSb NWs achieved by mixing different ratio of binary material powder sources. However the results are not consistent. For powder source ratio of 30:1 (the third highest InSb case), the lowest In concentration (9%) was obtained. The supporting information mentions that it “is probably due to the lower substrate employed for the optimized growth”. This is not clear and a more detailed explanation should be provided. In any case, this does not show a good and reproducible control of the NW growth/composition. Also, the paper claims that “... as shown in the typical SEM image of In_{0.28}Ga_{0.72}Sb NWs (Figure 1a), straight, smooth and dense NWs with the length greater than 10 um are non-epitaxially grown on the amorphous SiO₂/Si substrate. When the In concentration of the NWs is increased, the NW morphology is maintained with only the slight change in their growth density”. The NWs are clearly not straight and also there is an obvious density variation. Especially for the In_{0.15}Ga_{0.85}Sb NWs, the density is obviously lower. The description in the paper should be made more accurate. In fact some morphology change with different In concentration is expected and should be discussed with regard to the growth conditions.

Response:

- We thank for these valuable comments.
- Regarding the result inconsistency, we have repeated the experiments for many times and obtained the similar, consistent results. In particular, for the optimal growth condition of InGaSb NWs with around 9 % In content, the growth temperature is optimized at 505 °C, which is the lowest temperature required among all NWs with different In content. We have also tried the higher growth temperature (>505 °C) with the same powder mixing ratio, but this combination of growth parameters is out of the optimized process window grown with defective NWs with lots of surface coating. Since the growth temperature can drastically affect the chemical ratio of between Au and In in the AuIn alloy catalyst particle, which can subsequently affect the chemical composition of the grown NWs, we believe that the different growth temperature employed in this particular condition would contribute to the inconsistency here, where the powder source ratio of 30:1 (the third highest InSb case) yields the lowest In concentration (9%).
- In order to reinforce this standpoint, the following discussion: “**Note:** All associated values are extracted from more than 10 individual NWs for each sample group. It is noted that the powder source ratio of 30:1 would result in a relatively lower In concentration in In_xGa_{1-x}Sb NWs. This inconsistency should not be an accidental result as we have repeated the experiments for many times and obtained the similar results. For the optimal growth condition of InGaSb NWs with around 9 % In content, the growth temperature is optimized at 505 °C, which is the lowest temperature required among all NWs with different In content. We have also tried the higher growth temperature (>505 °C) with the same powder mixing ratio, but this combination of growth parameters is out of the optimized process window grown with defective NWs with lots of surface coating (Figure S2d). Since the growth temperature can drastically affect the chemical ratio of between Au and In in the AuIn alloy catalyst particle, which can subsequently

affect the chemical composition of the grown NWs, we believe that the different growth temperature employed in this particular condition would contribute to the inconsistency here, where the powder source ratio of 30:1 (the third highest InSb case) yields the lowest In concentration (9%).” is added to page 2 of the revised Supporting Information.

- Regarding the changes on the NW density, we have revised the description: “Furthermore, as shown in the SEM image of typical $\text{In}_{0.28}\text{Ga}_{0.72}\text{Sb}$ NWs (Figure 1a), smooth and clean NWs with the length greater than 10 μm are grown on the amorphous SiO_2/Si substrate. When the In concentration of the NWs is increased, the NW morphology is maintained, but there is a slight change on their growth density due to the increasing amount of InSb powder in the precursor source mixture. (Supporting Information Figure S2).” in order to clearly illustrate our argument in page 6 of the revised manuscript.

2. The chemical compositions of the NWs are obtained by EDS. The NW crystal structures are studied by TEM with very similar results. Surprisingly there is no mentioning of any defects such as stacking faults, for any of the NWs. I wonder how thorough the NWs were examined? The authors should clarify this. To get a better understanding of the NW properties and their corresponding device performance, optical spectroscopy such as photoluminescence should also be performed to measure the bandgap as well as the optical quality of the NWs.

Response:

- We appreciate for the input.
- Regarding the evaluation of crystal defects, we have in fact performed through HRTEM studies for 5 NWs for each NW composition (Figure 2, Figure S4, S5 and S6). We observed that there is not any noticeable 1D and 2D crystal defects such as stacking faults, twin boundaries and others, indicating the high crystal quality of the obtained NWs.
- For the additional characterization, we have measured the reflectance spectra of the NWs with different compositions, in which these data are added to Figure S11 and S12 in the revised Supporting Information. Also, more discussion: “Furthermore, reflectance spectra were taken to evaluate the optical properties of the NWs (Supporting Information Figure S11). As anticipated, the band gap of the NWs is found to decrease with the increasing In content (Supporting Information Figure S12), indicating the effective incorporation of In into the GaSb lattice as well as the good composition uniformity of the NWs.” is added to page 9 of the revised manuscript.

3. In this work, a very high hole mobility of 463 cm^2/Vs has been obtained. However for mobility calculation, the authors didn't show how to calculate the gate capacitance of the NW and what are the various parameters used. The Reference [5] cited didn't show the calculation either. In Table I the work by other researchers normally use 300 nm SiO_2 and metallic cylinder model to calculate gate capacitance, the work only use 50nm SiO_2 - it is important that it provides sufficient information to confirm the calculation and results.

Response:

- We thanks for the comments. In this work, the gate capacitance was calculated by the finite element method using COMSOL Multiphysics software. The model and calculated results are now given in Supporting Information Figure S14. In order to clearly illustrate this argument, the following discussion: “ C_{ox} is the gate capacitance and L is the channel length. C_{ox} can be

obtained from the finite element method by using COMSOL MultiPhysics software (Supporting Information Figure S14). For a typical $\text{In}_{0.09}\text{Ga}_{0.91}\text{Sb}$ NW transistor, when L is 4.2 μm , NW diameter is 41 nm, gate capacitance is determined to be 0.26 fF for 50 nm SiO_2 dielectric layer from COMSOL and peak transconductance is found to be 7×10^{-8} S for $V_{\text{ds}} = 0.1$ V (Supporting Information Figure S15), then the peak μ of the NW device can be calculated as high as 463 cm^2/Vs (Figure 4c). This mobility value is higher than the one of pure GaSb NWs.....” is added to page 14 of the revised manuscript.

4. It is shown that the 9% InGaSb NWs have the highest mobility and the results are presented in Fig. 4 (c) and (d). However Fig. 4(a) and (b) present the measurements of NW FET made of single $\text{In}_{0.28}\text{Ga}_{0.72}\text{Sb}$. Then the inset of Fig. 4(a) shows the SEM of $\text{In}_{0.09}\text{Ga}_{0.91}\text{Sb}$ NW. It is confusing and the paper should just provide all the mobility measurement results from $\text{In}_{0.09}\text{Ga}_{0.91}\text{Sb}$ NW FET to avoid confusion.

Response:

- We thank for pointing out our mistakes. All the results given in Figure 4a, b and c were for the $\text{In}_{0.09}\text{Ga}_{0.91}\text{Sb}$ NW device. We are sorry for the mistakes and we have corrected them in the revised manuscript.

5. For the NW photodetectors, despite that impressive performance has been achieved, there is not much comparison (among different type of NWs), discussion and explanation. For example, compared with visible, a much faster response has been obtained for IR detection. Why? Compared with the $\text{In}_{0.28}\text{Ga}_{0.72}\text{Sb}$ NWs, the $\text{In}_{0.09}\text{Ga}_{0.91}\text{Sb}$ NW seems to have much lower noise and larger photocurrent (Fig. S11). Why the paper claims that $\text{In}_{0.28}\text{Ga}_{0.72}\text{Sb}$ is the optimized device especially when the $\text{In}_{0.09}\text{Ga}_{0.91}\text{Sb}$ NWs have higher mobility? How about the other two types of NWs with intermediate In compositions? How do they behave (in terms of responsivity, response time and detectivity) and why? To better understand the photodetector performance of the NWs with different In concentration (and thus bandgap), the dark current and spectral response curves should also be provided to fully understand the device performance.

Response:

- We appreciate for these valuable comments.
- Since the response speed of a photodetector is mostly related to the carrier lifetime and carrier trap concentration, that is $\tau_0 = (1 + p_t/p)\tau$, where τ is the lifetime of photogenerated carriers, p_t is the trapped carrier density and p is the photo-generated free carrier density (A. Rose, *Concepts in Photoconductivity and Allied Problems*, Roberts Krieger Publishing Co., New York, 1978), the response speed should have an insignificant dependence on the detection wavelength. In this case, we have repeated the photoresponse measurement of the photodetector with a new 405 nm laser and a 532 nm laser used previously, which can now be modulated in a higher frequency (1 kHz) by directly tuning the power source of the laser, instead of using a mechanical shutter. The measurement result is then given in the revised Figure 5d. It is found that the rise and decay time constants are 39 and 46 μs , respectively for the 532 illumination (instead of in the range of milliseconds), which are consistent with the response observed in the IR region as well as being consistent with the theory discussed above.
- It should be pointed out that in our previous measurements in the visible range, the modulation frequency cannot be modulated in high frequency so that low frequency (e.g. 20 Hz) was used.

However, the low-frequency modulation is not the reason for obtaining the inaccurate response time if the mechanical shutter is fast enough. After a careful consideration, we have identified the problem, which is due to the inappropriate use of the low noise current amplifier. For example, if we employed a low-frequency low pass filter (30 Hz), the response curve can be affected drastically. However, if a higher frequency filter is used, the curve would become very noisy. In order to solve the problem, a higher modulation frequency should be utilized if the device response is fast enough. In this manner, we resolved the noise issue and obtained the accurate photoresponse of all devices here. More discussion of “This way, the rise and decay time constants are impressively found to be 39 and 46 μ s, respectively, indicating the fast response of the device.” is then added to page 17 of the revised manuscript in order to clarify the confusion.

- At the same time, the repeated measurement results of the InGaSb NW photodetector under 405 nm illumination with different Indium concentration are added to revised Supporting Information Figure S17, S19 and Table S3). Importantly, we compared the photocurrent (Figure S17 and S19), responsivity, detectivity and response time of the devices (Table S3), in which there is not any noticeable variation among all parameters as the In concentration changes, indicating the potency of these NWs for broadband photodetection in the visible regime. More discussion of “Photodetection performance of the NWs with other In contents were also measured and the results are shown in Supporting Information Figure S19 and Table S3. It is clear that there is not any significant difference in their performances among different NW compositions. Typically, the photocurrent is proportional to the product of carrier mobility and carrier lifetime ($I_p \propto \mu\tau$);⁴³ however, the density of different types of recombination centers can drastically change the value of carrier lifetimes here.⁴³ As a result, it is anticipated that those NWs would exhibit the similar photodetection performance although their carrier mobilities are different.” is added to page 17 and 18 of the revised manuscript.
- In addition, we have also carried out more thorough photoresponse measurements of the InGaSb NW devices under 1550 nm illumination and compiled all the results presented in Figure S20 and Table S4 of the revised Supporting Information. For example, we have added all the dark current characteristics in Figure S20 inset and there is not any noticeable difference on the dark current among different NW compositions. Similarly, as anticipated, NW devices with different In content exhibit the similar response time (in the order of tens of μ s), which agrees well with the theory discussed above. On the other hand, it is interesting that the NWs with the highest In content (i.e. In_{0.28}Ga_{0.72}Sb with the smallest bandgap here) yield the highest photoresponsivity, external quantum efficiency (EQE) and detectivity among all NWs studied. In contrast to the hot carriers effect observed for the photodetection in the visible region, the In_{0.28}Ga_{0.72}Sb NWs probably have the largest absorption coefficient towards 1550 nm irradiation although it has the relatively low mobility as compared to other In_xGa_{1-x}Sb NWs. Combined with the highest NW density obtained on the growth substrates (Figure 1 and Figure S2), the In_{0.28}Ga_{0.72}Sb NWs are concluded to be the optimized device channel material for the IR photodetection in this work.
- In order to clearly illustrate this argument, the following discussion of “the IR response of those NW devices would enhance substantially when the In concentration of the NW device channel is increased accordingly, which can probably be attributed to the reduced bandgap as

well as the enhanced absorption coefficient towards 1550 nm irradiation (Supporting Information Figure S11, S20 and Table S4)” is added to page 19 of the revised manuscript.

6. For the photodetectors, the photocurrents sometimes are shown in nA and sometimes are shown in μ A. I suppose for the former the dark current has been subtracted? For the comparison, it is better to show all the results (except for the response time measurements) under the same illumination intensity with dark current subtracted.

Response:

- We thank you for the good suggestion. We have subtracted the dark current in the revised manuscript as shown in the revised Figures 5 and 6, and all units of photocurrents are now changed to nA.

7. P. 22, it is mentioned that “the internal quantum efficiency is known to be far smaller than 1 because of the reflectance of light, insufficient absorption of light attributable to the finite thickness of NW and anisotropic absorption of light arising from the one-dimensional geometry”. This is not correct. Internal quantum efficiency (IQE) is defined as the ratio of the number of charge carriers that are generated to the number of photons that are absorbed. It is mainly related to the material quality rather than the reflectance of the light etc. The description above is more relevant to EQE.

Response:

- We appreciate for the comments and we are sorry for the inappropriate use of term of “internal quantum efficiency”. Here, it is suitable to refer it as “quantum efficiency”. The definition of quantum efficiency is the fraction of the incident optical power that contributes to electron-hole pair generation, which is widely used in photoconductive detectors (B. E. A. Saleh, M. C. Teich, *Fundamental of Photonics*, John Wiley & Sons, Inc., New York, chapter 17, P649). The definition is different from internal quantum efficiency (IQE) and external quantum efficiency (EQE). The ICE considers the fraction of light absorbed by the detector, while here it considers all the light incident on the detector. EQE is the ratio of the number of charge carriers collected by electrodes to the number of photons incident on the detector. In fact, the EQE is the product of quantum efficiency and gain. In order to clearly clarify this argument, we have changed the term of “internal quantum efficiency” to “quantum efficiency”, and the definition of the quantum efficiency is also given in page 23 of the revised manuscript.

8. I also suggest the following minor corrections:

1) Fig. 1(b) is not an EDS image, it is an SEM image.

2) Please give the full name of “TEM” in Page 6, “NP” in Page 10. In Page 3, please add (MBE) after “molecular beam epitaxy” since MBE is used in the text afterwards.

3) In Fig. 3a, better to use the same colors and order for Au, In, Sb, Ga as Fig. 3b. The current colors and order is confusing.

4) Please provide a reference for Equation 1 and also the values for V_{gs} , g_m , C_{ox} , and L .

5) For Equation 3, the definition of area “S” should be given.

6) In the manuscript, when discussing the photodetection performance, the single NW device is often referred as single NW FET. This could be confusing and I suggest to call it as single NW photodetector.

7) Page 19, “In addition, the good repeatability and fast response speed are also two important

requirement for advanced photodetectors”, “requirement” should be “requirements”.

Response:

- We thank you for pointing out all those mistakes. We have corrected all of them throughout the entire revised manuscript.

Reviewer #3:

1. The authors mentioned that the mobility distribution can be well fitted by Gauss function, however, no Gauss fitting was made in the manuscript.

Response:

- We thank for the comment. Gauss fittings were made and the fitting lines are now shown in **Figure 4d** of the revised manuscript.

Figure R2. Statistics of the extracted mobility of several NW FETs with different NW stoichiometry as the device channel.

2. The information given in Table S1 is confusing, especially the percentage combustion being inconsistent with the values given in the table. For example, for the ratio 20:1, the mass of source is 1 g, and source loss is 0.26 g; in this case, the combustion percentage should be 26%, instead of 25.1% given in the table. The authors should make all the calculation more explicit. Furthermore, it is not suitable to use the term “combustion percentage” because the source is not combusted during the synthesis.

Response:

- We thank for pointing out the mistakes. For clarity, as the term of “combustion percentage” is not an important parameter in this study, we decide to delete it in **Table S1** of the revised supporting information.

3. Similarly, for Table S2, why doesn't the molar ratio of In in the nanowires simply increase with the increasing of mass ratio of InSb powder? This reason would be important for the controllable synthesis of ternary III-V nanowires.

Response:

- We appreciate for the valuable input. In fact, as responded to reviewer#2 (question 1), we have repeated the experiments for many times and obtained the similar, consistent results. In particular, for the optimal growth condition of InGaSb NWs with around 9 % In content, the

growth temperature is optimized at 505 °C, which is the lowest temperature required among all NWs with different In content. We have also tried the higher growth temperature (>505 °C) with the same powder mixing ratio, but this combination of growth parameters is out of the optimized process window grown with defective NWs with lots of surface coating. Since the growth temperature can drastically affect the chemical ratio of between Au and In in the AuIn alloy catalyst particle, which can subsequently affect the chemical composition of the grown NWs, we believe that the different growth temperature employed in this particular condition would contribute to the inconsistency here, where the powder source ratio of 30:1 (the third highest InSb case) yields the lowest In concentration (9%).

- In order to reinforce this standpoint, the following discussion: “**Note: All associated values are extracted from more than 10 individual NWs for each sample group. It is noted that the powder source ratio of 30:1 would result in a relatively lower In concentration in $\text{In}_x\text{Ga}_{1-x}\text{Sb}$ NWs. This inconsistency should not be an accidental result as we have repeated the experiments for many times and obtained the similar results. For the optimal growth condition of InGaSb NWs with around 9 % In content, the growth temperature is optimized at 505 °C, which is the lowest temperature required among all NWs with different In content. We have also tried the higher growth temperature (>505 °C) with the same powder mixing ratio, but this combination of growth parameters is out of the optimized process window grown with defective NWs with lots of surface coating (Figure S2d). Since the growth temperature can drastically affect the chemical ratio of between Au and In in the AuIn alloy catalyst particle, which can subsequently affect the chemical composition of the grown NWs, we believe that the different growth temperature employed in this particular condition would contribute to the inconsistency here, where the powder source ratio of 30:1 (the third highest InSb case) yields the lowest In concentration (9%).**” is added to page 2 of the revised Supporting Information.

4. Regarding the nanowire array photodetectors, the reduced R and D^ should not arise from the higher parasitic capacitance. The capacitance can only affect the response speed of a photodetector. While the speed of the nanowire array photodetector is almost the same with single nanowire device as presented by the authors, this indicates that there is not any higher parasitic capacitance existed for the nanowire array devices.*

Response:

- We thank for pointing out the mistakes. The quality of the printed NW arrays may contribute to the main reason for the reduced R and D^* because of the improper alignment of the NW arrays, NW-to-NW variations, breakage of NWs as well as non-optimized electrical contact between NWs and electrodes. In this regard, we have modified the discussion of “...which is probably due to the improper alignment of the NW arrays, NW-to-NW variations, breakage of NWs as well as non-optimized electrical contact between NWs and electrodes” in page 22 of the revised manuscript.

List of Other Changes:

1. The Figure 4 caption is revised as the new inset is added to the figure.
2. The figure sequence in Supporting Information is rearranged as new figures are added.
3. The following discussion of “**It is still controversial whether the epitaxial relation between the**

NW and the catalyst particle is caused by a quasi-vapor-solid-solid process or vapor-liquid-solid (VLS) process.³¹” is added to page 8 of the revised manuscript.

4. The following discussion of “It should also be pointed out that the growth of ternary $\text{In}_x\text{Ga}_{1-x}\text{Sb}$ NWs is different from the one of GaSb NWs as reported in our previous work,⁵ where AuGa alloy droplets was employed as the catalyst there. Furthermore, the ternary NWs show a smaller diameter as compared with typical GaSb NWs. The addition of In can affect the NW growth dynamics while the uncontrolled radical NW growth may be constrained.” is added to page 12 of the revised manuscript.

Reviewers' comments:

Reviewer #1 (Remarks to the Author):

The authors for the manuscript " Ultra-Fast Photodetectors based on High-Mobility InGaSb Nanowires" have done a good job in answering the questions and points raised in the first round of the review. However, even so, my main concern about this manuscript is that the results not are surprising or remarkable. This impression has not changed during the review. One of the highlights in the manuscript is that the hole mobility of GaSb can be substantially increased by the introduction of 9% In and that the increase decreases with further introduction of In, as demonstrated in fig 4. At the same time, it is clear from statistics that there is a large spread in the data and that the max value used for benchmarking is cherry-picking among the devices, as evident from Fig 4d. However, part of this story is that the reference values for GaSb are comparably low and the apparent increase in mobility might be a particular characteristics of the particular material synthesis used. One further particularity is that the 9% devices can very efficiently be turned off by the use of a back-gated transistor structure. It is very surprising and not commonly seen for GaSb nanowires. The observation is a hint that surface modifications may play a crucial role in the change in hole mobility.

Although the investigation is valuable and will find interest in the community, I do not recommend it for publication in Nature Communications. My judgement is based on vague surprise-factor, remaining questions on the mobility values, and a suspicion that the determined hole mobility not only is an effect of the In concentration as claimed in the manuscript, but rather an effect of surface and possible point defect modification related to the specific growth method used.

Reviewer #2 (Remarks to the Author):

The authors have carefully addressed the comments from all the reviewers and provided additional results/information to support the manuscript. I think the quality of the revised manuscript has been significantly improved and is now suitable for publication in Nature Communications.

Reviewer #3 (Remarks to the Author):

The authors have addressed all my questions, please accept it as is.

Response to Referees' Comments and Suggestions
on manuscript NCOMMS-18-11449A

We appreciate the referees for considering our manuscript and providing valuable comments. Accordingly, changes have been made in the manuscript, highlighted with red color. Below is our response to reviewers' comments.

Reviewer #1

The authors for the manuscript "Ultra-Fast Photodetectors based on High-Mobility InGaSb Nanowires" have done a good job in answering the questions and points raised in the first round of the review. However, even so, my main concern about this manuscript is that the results not are surprising or remarkable. This impression has not changed during the review. One of the highlights in the manuscript is that the hole mobility of GaSb can be substantially increased by the introduction of 9% In and that the increase decreases with further introduction of In, as demonstrated in fig 4. At the same time, it is clear from statistics that there is a large spread in the data and that the max value used for benchmarking is cherry-picking among the devices, as evident from Fig 4d. However, part of this story is that the reference values for GaSb are comparably low and the apparent increase in mobility might be a particular characteristics of the particular material synthesis used. One further particularity is that the 9% devices can very efficiently be turned off by the use of a back-gated transistor structure. It is very surprising and not commonly seen for GaSb nanowires. The observation is a hint that surface modifications may play a crucial role in the change in hole mobility. Although the investigation is valuable and will find interest in the community, I do not recommend it for publication in Nature Communications. My judgment is based on vague surprise-factor, remaining questions on the mobility values, and a suspicion that the determined hole mobility not only is an effect of the In concentration as claimed in the manuscript, but rather an effect of surface and possible point defect modification related to the specific growth method used.

Although the investigation is valuable and will find interest in the community, I do not recommend it for publication in Nature Communications. My judgement is based on

vague surprise-factor, remaining questions on the mobility values, and a suspicion that the determined hole mobility not only is an effect of the In concentration as claimed in the manuscript, but rather an effect of surface and possible point defect modification related to the specific growth method used.

Response:

- We appreciate for the valuable comments.
- In fact, as shown in Figure 4d, the mobility data of all $\text{In}_x\text{Ga}_{1-x}\text{Sb}$ nanowire transistors has indeed the similar and wide distributions, while the mobility distribution is relatively narrow for pure GaSb nanowire devices. Usually, the calculated mobility values are affected by many factors, such as the diameter of nanowires, crystallinity of nanowires, electrical contact quality, and surface contamination of nanowire channels during device fabrication, etc. Those factors cannot be the same for each devices, which leads to the spread of the mobility distribution of GaSb nanowire transistors. For $\text{In}_x\text{Ga}_{1-x}\text{Sb}$ nanowires, the variation of both In and Ga concentration (Supplementary Information Table S2) in each nanowire sample groups would give the additional variation to the corresponding mobility values, contributing to the wider mobility distribution as compared to pure GaSb nanowire devices. In order to ensure the accuracy of the mobility data of all nanowire transistors, more GaSb and $\text{In}_x\text{Ga}_{1-x}\text{Sb}$ nanowire devices have been fabricated and then their motility values are evaluated as presented in the **revised Figure 4d**. It is clear that the average and maximum mobility values stay relatively the same when the sampling size is increased from 20 devices to 40 devices for each nanowire sample groups. In any case, we did not cherry-pick the maximum mobility values of our nanowire devices to benchmark against other III-V nanowire device mobility values reported in the literature. Actually, when Figure 4c inset was compiled, only the maximum mobility values of various single nanowire devices were compared, including our $\text{In}_{0.09}\text{Ga}_{0.91}\text{Sb}$ nanowires studied here, p-type carbon-doped InSb and zinc-doped GaAs nanowires, etc. In addition, the average mobility values of these different single nanowire devices are currently also compiled and depicted in **Supplementary Information Figure S15c**. Both

figures demonstrate evidently that the mobility of our $\text{In}_{0.09}\text{Ga}_{0.91}\text{Sb}$ nanowire devices (regardless of the maximum and average values) are superior among many others of III-V nanowire devices reported till now.

- At the same time, in order to investigate the origin of the relatively low mobility of obtained GaSb nanowires, high-resolution transmission electron microscopy (HRTEM) measurements were carried out. As shown in the typical HRTEM image in Supplementary Information Figure S16e and f, it is obvious that there are many lattice defects (e.g. stacking faults and inversion domains, etc) existed in the GaSb nanowire, which is in a distinct contrast to the ones of $\text{In}_x\text{Ga}_{1-x}\text{Sb}$ nanowires as presented in Figure 2 and Supplementary Information Figure S4 to S6. These defects could result from the notorious surfactant effect of Sb constituents during the nanowire growth (*Nat. Commun.*, **2014**, 5, 5249). These large amounts of defects could contribute to the relatively low mobility of GaSb nanowires due to the severe carrier scattering there. Furthermore, these defects can also provide a large amount of free carriers that cause the down shift of the Fermi level of GaSb to its valance band maximum. As a result, it is difficult to deplete the free carriers of the nanowire to achieve the device OFF state by electrical back-gating. It is also a common phenomenon observed for back-gated GaSb nanowire transistors that cannot be turned off at room temperature (*Appl. Phys. Lett.*, **2011**, 99, 262104; *RSC Adv.*, **2013**, 3, 19834). On the other hand, the minimized lattice defect concentration of $\text{In}_x\text{Ga}_{1-x}\text{Sb}$ nanowires could contribute to the correspondingly enhanced device carrier mobility. Moreover, even utilizing sulfur surfactants during the growth to obtain highly crystalline GaSb nanowires, the maximum device mobility value of $400 \text{ cm}^2\text{V}^{-1}\text{s}^{-1}$ (*ACS Nano*, **2015**, 9, 9268) is still lower than the one of $\text{In}_{0.09}\text{Ga}_{0.91}\text{Sb}$ nanowire devices with the comparable nanowire diameter, which suggests the introduction of In in GaSb nanowires not only reduces the defect concentration but also leads to the favorable changes in their electrical device performance. As a result, the device mobility values of $\text{In}_x\text{Ga}_{1-x}\text{Sb}$ nanowires are much improved as compared to the pure GaSb nanowire ones with the nanowire channel grown by the similar technique.

- Furthermore, the mobility values of nanowire devices can also be affected by the channel surface condition, such as the surface roughness, passivation and adsorbents, etc. In this work, since both GaSb and InGaSb nanowires have the smooth surface as revealed from the TEM characterization in Figure 2, Supplementary Information Figure S4 to S6 and S16, the surface roughness effect should be the same among all these nanowires. To further exclude the effect of any surface modification (e.g. adsorbents) on the device mobility of the nanowires, we measured the transfer curves of a typical $\text{In}_{0.09}\text{Ga}_{0.91}\text{Sb}$ nanowire transistor in both air and vacuum (3.5×10^{-4} Pa) as shown in **Supplementary Information Figure S17**. Explicitly, there is not any noticeable difference observed for the measurement result in both air and vacuum, which suggests that the adsorbents have insignificant effect on the electrical properties of nanowire devices. Then, surface passivation with ammonia sulfide $(\text{NH}_4)_2\text{S}$ were also performed on the $\text{In}_{0.09}\text{Ga}_{0.91}\text{Sb}$ nanowire transistor. As presented in **Supplementary Information Figure S18**, the surface passivation can slightly reduce the subthreshold swing and move the subthreshold voltage towards the negative voltage direction by minimizing the surface trap concentration on the nanowire device channel, which can further enhance the corresponding device mobility to some extent. This phenomenon indicates that the initial high device mobility values of our $\text{In}_{0.09}\text{Ga}_{0.91}\text{Sb}$ nanowires are indeed attributed to their intrinsic material properties, instead of relating to any surface passivation effect. In this case, the effective electrical back-gating to efficiently turning the device ON and OFF can also be attributed to the intrinsic nanowire properties, rather than any other extrinsic effect (e.g. surface modification).
- In this regard, in order to reinforce all the above argument that the enhanced device mobility of $\text{In}_{0.09}\text{Ga}_{0.91}\text{Sb}$ nanowires is mainly attributed to both the effective incorporation of In into the nanowire lattice and the minimized lattice defect concentration, instead of relating to any surface modification effect, **we have included all these discussion in page 13, 14 and 31 of the revised manuscript as well as added Figure S15c, S16e, S16f, S17 and S18 in the revised Supplementary Information.**

Reviewer #2

The authors have carefully addressed the comments from all the reviewers and provided additional results/information to support the manuscript. I think the quality of the revised manuscript has been significantly improved and is now suitable for publication in Nature Communications.

Response:

- We thank for the appreciation from the reviewer.

Reviewer #3

The authors have addressed all my questions, please accept it as is.

Response:

- We thank for the appreciation from the reviewer.

Other List of Change:

1. In page 2, **the abstract is shortened to less than 150 words.**
2. In page 5 (Figure 1), page 7 (Figure 2), page 9 (Figure 3) and page 11 (Figure 4), the scale bars are not labelled within the figure. **The length of all scale bars is defined in the legend in page 31.**
3. In page 11, Figure 4d, **the use of green color is avoided.**
4. In page 13, the sentence is revised to the following: **“this hole mobility value is also compared and contrasted with other maximum mobility values reported for III-V NWs in the literature.”**
5. In page 13, “20 NW devices” are changed to **“40 NW devices”**.
6. **The location of Table 1 is moved to page 24.**
7. In page 26, the statement of **“The source data is provided as a Source Data file.”** is added and the statement of **“author contributions”** is updated.
8. In page 31, the Figure 4 legend is revised to the following: **“.....The maximum hole mobility of InAs, InSb, C-doped InSb, Zn-doped GaAs, GaSb, and Zn-doped GaSb is adapted from reference ³³, ³⁴, ³⁵, ³⁶, 4, and ³⁷. Their average hole mobility**

values are also compiled in Supporting Information Figure S15c. (d) Statistics of the extracted mobility of ~40 NW FETs with different NW stoichiometry as the device channel.”

9. All legends are displayed underneath the table.
10. All figure and table legends are moved to the end of the revised manuscript.
11. All unit dimensions are expressed using negative integers throughout the entire revised manuscript.
12. All the phrases of “Supporting Information” are changed to “Supplementary” throughout the entire revised manuscript.
13. Since new figures are added to the revised Supplementary Information, the sequence of all the figures are all re-arranged.